# DEER: A Benchmark for Evaluating Deep Research Agents on Expert Report Generation

Janghoon Han [1]  Heegyu Kim [1]  Changho Lee [1]  Dahm Lee [1]  Min Hyung Park [1]  Hosung Song [1]
Stanley Jungkyu Choi [1]  Moontae Lee [1 2]  Honglak Lee [1 3]

## Abstract

Recent advances in large language models have enabled deep research systems that generate expert-level reports through multi-step reasoning and evidence-based synthesis. However, evaluating such reports remains challenging: report quality is multifaceted, making it difficult to determine what to assess and which criteria to use; LLM-based judges may miss errors that require domain expertise to identify; and because deep research relies on retrieved evidence, report-wide claim verification is also necessary. To address these issues, we propose DEER, a benchmark for evaluating expert-level deep research reports. DEER systematizes evaluation criteria with an expert-developed taxonomy (7 dimensions, 25 subdimensions) operationalized as 101 fine-grained rubric items. We also provide task-specific Expert Evaluation Guidance to support LLM-based judging. In addition to rubric-based assessment, we propose a claim verification architecture that verifies both cited and uncited claims and quantifies evidence quality. Experiments show that current systems produce structurally plausible, evidence-citing reports, but still struggle to fully satisfy expert-level user requests and achieve logical completeness. Beyond performance comparisons, DEER makes system strengths and limitations interpretable and provides diagnostic signals for improvement.[1]

## 1. Introduction

Driven by rapid advances in large language models (LLMs), automated deep research systems are emerging as a core

[1]LG AI Research [2]University of Illinois Chicago [3]University of Michigan, Ann Arbor. Correspondence to: Janghoon Han <hanjh9439@gmail.com>.

*Proceedings of the 43rd International Conference on Machine Learning, Seoul, South Korea. PMLR 306, 2026. Copyright 2026 by the author(s).*

[1]Code and data: github.com/hanjanghoon/DEER.

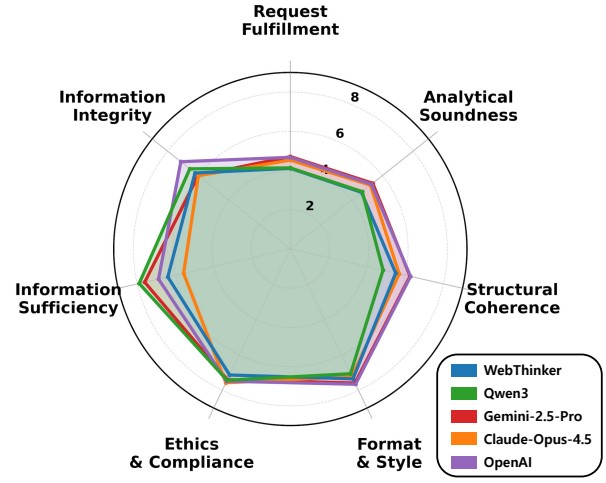

*Figure 1.* Deep Research System Performance Comparison. Results for five systems on the proposed benchmark.

technology in both academia and industry (OpenAI, 2025a; Google, 2025; Anthropic, 2025; Yang et al., 2025; Li et al., 2025b; Huang et al., 2025; Li et al., 2026). Unlike conventional web search, these systems address complex research queries by decomposing them into multiple steps and dynamically seeking additional information based on intermediate results. Through this process, they integrate information from diverse sources and synthesize multiple perspectives to produce reliable, evidence-based research reports (Xu & Peng, 2025; Zhang et al., 2025; Java et al., 2026). As a result, deep research systems can achieve strong performance even on challenging benchmark tasks (Mialon et al., 2024; Phan et al., 2025).

Early evaluations of deep research systems relied primarily on complex reasoning benchmarks (Rein et al., 2024; Mialon et al., 2024; Phan et al., 2025), which indirectly assessed information gathering, hypothesis testing, and multi-step reasoning through task performance. Subsequently, deep web search QA benchmarks were introduced to more directly measure systems' web browsing and information retrieval abilities—core capabilities of deep research—by evaluating multi-step search, information integration, and answer derivation (Wei et al., 2025a; Krishna et al., 2025; Mialon et al., 2024; Chen et al., 2025; Gou et al., 2026). More

recently, deep research report benchmarks have emerged to evaluate the quality of generated reports from multiple perspectives, moving beyond simple short-answer-based evaluation (Consult, 2025; Coelho et al., 2025; Du et al., 2026; Wan et al., 2026).

Despite these advancements, existing evaluation methods for deep research systems continue to face significant limitations when applied to expert-level reports. First, evaluation criteria are often underspecified, leaving it unclear what aspects of report quality should be assessed. Prior benchmarks typically evaluate reports using coarse, high-level dimensions, and thus do not provide sufficiently fine-grained criteria for precise assessment (Consult, 2025; Coelho et al., 2025). Moreover, even when fine-grained evaluation items are introduced, they are often generated or structured by LLMs, which can undermine consistency and reliability (Du et al., 2026; Wan et al., 2026; Wang et al., 2026). Second, even with well-specified rubric items, evaluations that rely on LLM judges may fail to identify issues that require domain expertise. Third, current approaches to source verification are typically restricted to claims with explicit citation markers, leaving factual reliability across the full report insufficiently examined. Together, these limitations hinder comprehensive and reliable evaluation of deep research systems.

To address these limitations, we propose DEER (the **DE**ep research **E**xpert **R**eport benchmark), which evaluates deep research reports through 50 report-generation tasks spanning 13 domains. DEER surveys established reporting norms and evaluation criteria across domains and synthesizes them, through an expert consensus process, into a Deep Research Report Evaluation Taxonomy comprising 7 dimensions and 25 subdimensions. Based on this taxonomy, DEER evaluates each report across two complementary components: (i) report-quality assessment, which requires holistic, document-level judgment, and (ii) external-information verification, which can be assessed at the claim level by checking against external sources. For report quality, we operationalize the taxonomy into a fixed set of 101 fine-grained rubric items and supplement them with task-specific evaluation guidance authored by domain experts to improve the consistency and validity of LLM-based scoring. For external information, we introduce an information-verification module that examines both cited and uncited claims across the report and produces quantitative measures of evidence quality. DEER then integrates rubric-based scores with these metric-based signals to yield a unified, multidimensional assessment of each expert report. Figure 1 presents an overview of dimension-level performance across deep research systems evaluated using DEER.

The main contributions of this work are as follows:

- We present DEER, a systematic and interpretable

benchmark for evaluating deep research reports, grounded in a hierarchical evaluation taxonomy.

- We translate the taxonomy into a standardized set of 101 fine-grained rubric items and provide task-specific Expert Evaluation Guidance to support consistent and reliable LLM-based report scoring.

- We propose a report-level information-verification architecture that backtracks inter-claim dependencies to retrieve citations and verify claims, enabling more complete evaluation of claim reliability across entire reports.

## 2. Related Works

With the proliferation of high-performing LLMs, LLM-as-a-Judge approaches—using LLMs as evaluators—have been widely proposed and studied (Zheng et al., 2023; Liu et al., 2023; Chiang et al., 2024; Kim et al., 2024a;b). In this line of work, early evaluations of deep research systems primarily focused on how well models solve expert-level, high-difficulty questions (Rein et al., 2024; Phan et al., 2025). Subsequently, to directly assess a core property of deep research—accessing and leveraging external information from the open web—benchmarks were proposed to measure models' ability to search and browse the web, synthesize the required information, and construct answers grounded in that evidence (Gou et al., 2026; Wei et al., 2025a; Huang et al., 2025). More recently, beyond simple short-answer evaluation, studies have sought to evaluate the ability to generate long-form reports that require expert-level analysis, reasoning, and interpretation by integrating information from multiple sources (Consult, 2025; Coelho et al., 2025; Du et al., 2026; Wan et al., 2026; Wang et al., 2026; Yao et al., 2025; Sharma et al., 2026).

Despite this progress, existing report-evaluation methods are not based on expert-defined, fine-grained criteria. Some approaches rely on only a few coarse axes, leaving judgments largely to the evaluator LLM's implicit standards (Consult, 2025; Coelho et al., 2025). Even when rubrics are subdivided, their specification and application can still depend substantially on the evaluator LLM (Du et al., 2026; Wan et al., 2026), making it unclear whether the resulting scores correctly reflect report quality or align with expert assessment. In addition, source verification is often omitted (Consult, 2025; Coelho et al., 2025) or limited to a subset of cited claims (Du et al., 2026; Wan et al., 2026), which may be insufficient to assess report-level factuality. To address these limitations, we propose an evaluation framework that (i) scores long-form research reports along multiple dimensions using expert-systematized, fine-grained criteria, and (ii) performs systematic source verification for claims

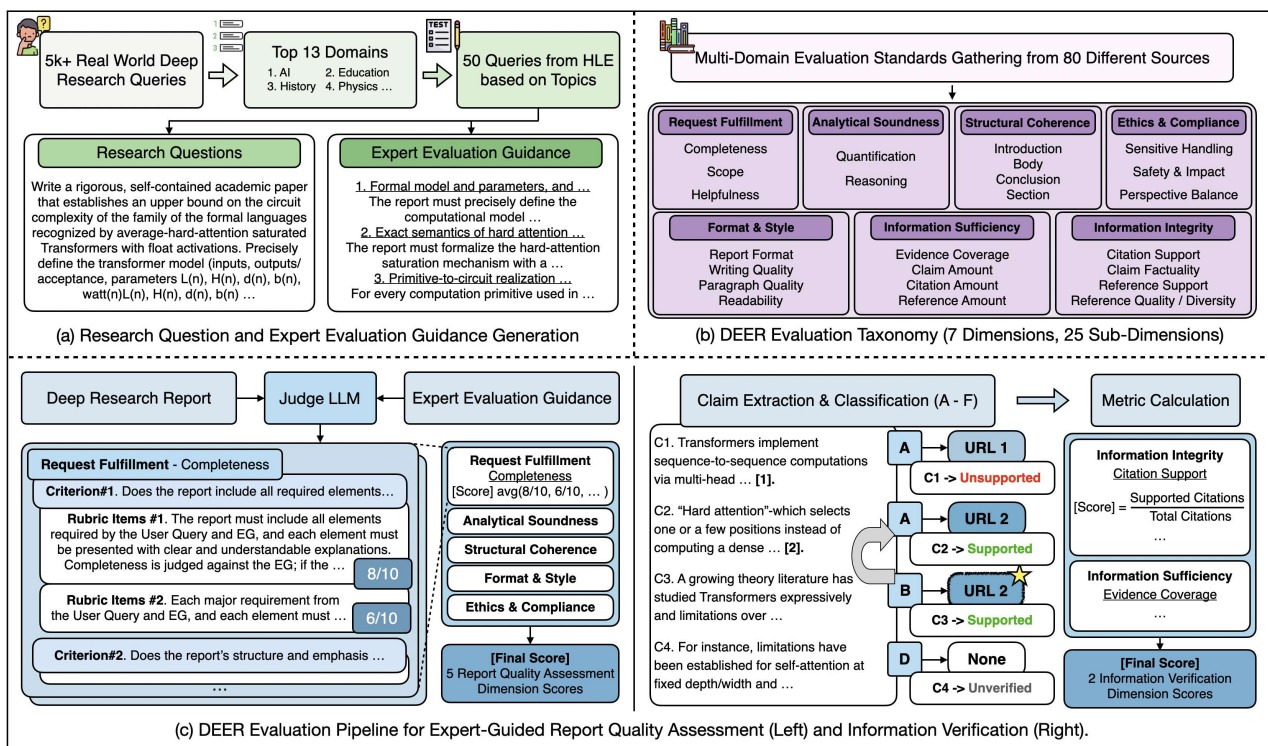

*Figure 2.* Overview of the DEER evaluation framework. (a) Research question and expert guidance generation from real-world deep research queries. (b) Construction of the Deep Research Evaluation Taxonomy consisting of 7 dimensions, 25 sub-dimensions, and 101 granular rubrics. (c) The DEER evaluation pipeline, which combines expert-guided LLM-as-a-judge scoring and claim-level information verification to assess deep research reports.

throughout the report. A detailed comparison with prior work is provided in Appendix A.

## 3. Data Construction

To construct an evaluation dataset that reflects real-world usage, we analyze 5,842 in-house user queries collected from an internal deep research system and derive a topic distribution. For topic classification, we adopt the taxonomy of Wettig et al. (2025), consistent with prior work (Du et al., 2026).

We use Humanity's Last Exam (HLE) (Phan et al., 2025) as the source of seed questions because it provides expert-written, high-difficulty, multidisciplinary questions that align well with the expert-level topics addressed in deep research reports. To match our target topic distribution, we map our topics to the 13 HLE subject domains and sample 50 seed questions accordingly.

Because HLE items are posed in a QA format, they need to be reformulated before they can be used as deep research report-generation queries. We therefore ask domain experts (each holding at least a master's degree or possessing equivalent expertise in the relevant field) to review the original questions, answers, and rationales to identify the underlying concepts, theories, and phenomena, and to reformulate each

item as a report- or paper-style prompt. Each reformulated prompt is drafted by one domain expert and cross-reviewed by another expert from the same field, with iterative revisions conducted as needed. During this process, we remove answer-revealing elements (e.g., factual conclusions, proofs, or specific answers) and retain only high-level writing directions, such as the intended scope of analysis and comparative perspectives. As a result, models are required to develop their reasoning and narrative independently. This reformulation shifts the task from producing a short answer to generating a report that requires expert-level analysis, reasoning, and interpretation. Detailed examples and procedures are provided in Appendix B.

## 4. Approach

### 4.1. Overview

To systematically evaluate deep research reports, we establish an evaluation framework grounded in two core aspects of deep research: external evidence acquisition and report-level synthesis (Java et al., 2026; Xu & Peng, 2025; Zhang et al., 2025). Based on these aspects, we construct a Deep Research Report Evaluation Taxonomy comprising seven major dimensions (§ 4.2), grouped into five report-quality dimensions and two external-information dimensions. The

report-quality dimensions focus on synthesis and presentation through holistic, document-level judgment, whereas the external-information dimensions assess how external evidence is acquired and used via claim-level verification.

Given these differences in evaluation characteristics, we propose a hybrid evaluation architecture that integrates methodologies tailored to each component, as illustrated in Figure 2. Expert-Guided Report Quality Assessment (§ 4.3) adopts an LLM-as-a-judge approach, combining a fixed set of granular rubric items with report-specific Expert Evaluation Guidance authored by domain experts to assess report quality against expert-designed criteria. Information Verification (§ 4.4) performs metric-based evaluation by automatically extracting claims and citations, then verifying claims against external sources, yielding quantitative measures of the sufficiency and integrity of external evidence. Scores from the rubric-based assessment and information-verification metrics are aggregated into seven dimension-level scores, which are then combined into an overall report score.

## 4.2. Deep Research Report Evaluation Taxonomy

In the absence of systematic evaluation criteria for deep research reports, we construct an evaluation taxonomy by synthesizing expert report assessment standards from multiple fields. We draw on 80 established standards across 20 domains of expertise, including systematic research reporting guidelines, technical and professional report-writing and evaluation guidelines, and academic publishing norms. A panel of experts with experience in deep research system development and academic reviewing iteratively analyzes and consolidates these standards to derive report-quality dimensions. We then introduce complementary external-information dimensions that reflect deep research-specific characteristics of external evidence use and verification, resulting in a taxonomy comprising seven major dimensions and 25 detailed subdimensions in total (Table 10). We validate the taxonomy through independent review by 10 experts from diverse domains. This hierarchical taxonomy organizes diverse quality elements into structured evaluation dimensions and subdimensions, enabling systematic, multidimensional assessment of deep research reports. It supports diagnosis of report strengths and weaknesses and provides a foundation for targeted improvement. Full descriptions, validation procedures, and the mapping from source standards to our criteria are provided in Appendix C.

## 4.3. Expert-Guided Report Quality Assessment

Existing evaluation methods exhibit two key limitations when applied to long-form expert reports. First, they typically apply broad, high-level criteria, granting LLM judges substantial discretion over which aspects of a report to attend to. As a result, judges may (i) examine only parts of a

report, (ii) consider only a subset of the many sub-aspects implied by each criterion, and (iii) focus on non-critical or superficial elements, leading to increased variance across judges and reduced evaluation consistency (Li et al., 2025b; Consult, 2025; Coelho et al., 2025). Second, even when the evaluation focus is clearly defined, assessing correctness and completeness often requires domain expertise that LLM judges may lack. Consequently, they may miss subtle but important issues, including non-obvious logical leaps, domain-specific misinterpretations, and fine-grained inaccuracies (Du et al., 2026). To address these limitations, this study (1) makes evaluation criteria explicit through a fixed set of fine-grained rubric items and (2) provides task-specific Expert Evaluation Guidance that includes domain-specific context and reference points, enabling LLM judges to identify hard-to-detect errors and omissions.

**Granular Rubric Design.** Broad evaluation criteria can lead to inconsistent LLM-based scoring. Recent work addresses this issue by decomposing high-level criteria into more granular rubrics (Lee et al., 2025; Wei et al., 2025b; Ruan et al., 2026). For deep research report evaluation, Du et al. (2026) follows a related approach by using an LLM to generate task-specific evaluation criteria. While this strategy narrows the evaluation focus, it raises two important concerns: whether the set of generated criteria is sufficiently comprehensive and captures what matters most in expert reports, and whether dynamically varying criteria enable interpretable and comparable scores that support systematic diagnosis of system strengths and weaknesses across tasks.

This study uses a fixed, expert-designed rubric to ensure reliable and interpretable evaluation. Building on the taxonomy in Table 10, a panel of experts operationalizes the 25 subdimensions into 46 evaluation criteria and translates each criterion into concrete, checkable rubric items. The items are organized into two aspects: coverage, whether required components are present and fully addressed wherever they occur in the report; and quality, the degree to which the targeted components are executed to a high standard. This process yields 101 scorable rubric items in total.[2] The rubric is applied identically to all deep research reports across 50 tasks, enabling diagnosis of the system's overall strengths and weaknesses with richer, more interpretable signals (see Appendix D for the rubric structure and examples).

**Expert Evaluation Guidance.** Even with a well-specified, fine-grained rubric, evaluating expert-level reports often requires substantial domain knowledge. In such settings, non-expert evaluators—including LLM judges—may fail to detect subtle domain-specific errors or omissions that subject-matter experts would identify. To mitigate this risk, we introduce task-specific Expert Evaluation Guidance, which

---

[2] The 46 criteria were validated through independent review by 10 experts following the procedure in § 4.2.

| Pos. | Sentence / Extracted Claim | Class |
|------|----------------------------|-------|
| L1.S1 | Multi-junction solar cells achieved efficiencies above 45% in laboratories [1]. | A (Explicit) |
| L1.S2 | This dramatic increase is due to the layering of different semiconductor materials. | B (Implicit → L1.S1) |
| L2.S1 | The enhanced efficiency will reduce the land area required for solar farms. | C (Cross-section) |
| L2.S2 | These new panels are durable enough to withstand a Category 4 hurricane. | F (No source) |

*Table 1.* An example of claim typing and back-tracking. Position labels follow the format Lx.Sy, where L denotes the paragraph index and S the sentence index within that paragraph. L1.S2 (Type B) has no explicit citation but inherits [1] from L1.S2 through back-tracking; L2.S2 (Type F) has no recoverable source."

provides domain-grounded context and reference points to support consistent and informed evaluation.

Expert Evaluation Guidance enumerates the required content elements and expert expectations for each task as concrete, verifiable statements that can be checked directly against the report. The guidance for each task is produced using the same expert drafting and cross-review procedure as the query reformulation process described in Section 3. Specifically, a domain expert reformulates each HLE item into a report-style prompt (Section 3). The expert then derives the guidance by identifying the substantive content an adequate expert report should cover under the task prompt. Each required element is expressed as a concrete, verifiable statement that can be checked against the report. The resulting guidance is subsequently cross-reviewed by another expert from the same field to identify omissions, redundancies, or ambiguities, and revised as needed. Through this process, Expert Evaluation Guidance complements the fixed rubric by anchoring evaluation in task-specific expert knowledge while maintaining consistency and interpretability across tasks. Appendix B.4 provides detailed guidance construction procedures and illustrative examples.

### 4.4. Information Verification

To evaluate the external-information dimensions (Information Integrity and Information Sufficiency) in a reproducible way, we verify claims against evidence across the entire report and summarize the results as quantitative metrics, rather than relying on free-form LLM scoring. Specifically, our module (i) extracts atomic claims and links each verifiable claim to the sources it should be checked against (including uncited claims via implicit-citation recovery), and (ii) verifies whether the linked evidence supports each claim and aggregates the outcomes into Integrity and Sufficiency metrics. Implementation details and metrics are presented in Appendix F.

**Claim Types and Verification Scope.** Existing citation checkers typically verify only explicitly cited sentences, excluding claims without citation markers from verification. However, because reports do not repeat citations for every claim, relying solely on citation markers may miss claims that are still verifiable against external sources. For example,

a claim may rely on a source cited earlier in the report, in which case its supporting evidence can be recovered from context even without an inline citation marker. To address this issue, we classify extracted claims into six types (A–F) and apply external-evidence verification only to the verifiable types (A–C): (A) inline-cited claims; (B/C) uncited claims whose support is available elsewhere in the report; and (D–F) structural, internal, or otherwise non-verifiable claims. Table 14 provides definitions and annotation guidance for each type, and Figure 9 shows the full classification prompt.

**Implicit Claim Back-Tracking.** A key challenge is that expert reports often contain implicit claims whose supporting citations appear in earlier sentences (or even earlier sections), so the claim itself has no explicit marker. To verify such claims, we introduce a semantic Back-Tracking mechanism that recovers the citation set needed for verification. For a sentence $s_i$, the LLM identifies the set of preceding sentences $R(s_i)$ that $s_i$ semantically depends on, and defines the valid citation set used for verification as:

$$\mathcal{V}(s_i) = \mathcal{C}(s_i) \cup \bigcup_{k \in R(s_i)} \mathcal{C}(s_k), \qquad (1)$$

where $\mathcal{C}(s_i)$ denotes the explicit citations of $s_i$. This enables verification by inheriting citations from previously referenced sentences, even when $\mathcal{C}(s_i) = \emptyset$, substantially expanding verification coverage beyond explicitly cited sentences. Table 1 shows a concrete example: the implicit sentence L1.S2 has no explicit citation but semantically depends on the preceding sentence L1.S1, so back-tracking inherits citation [1] from L1.S1 to verify L2.S1. We empirically validate this mechanism against expert-annotated dependencies in §6.5 and Appendix F.2 (Table 15). Moreover, the pipeline is robust to mispredicted dependencies because inherited citations serve only as candidate evidence. A claim is marked as supported only when the subsequent verification stage confirms that the candidate evidence actually supports it; otherwise, it is marked as "not supported."

**Evidence-Grounded Verification and Metrics.** For each verifiable claim (Types A–C), we retrieve the evidence documents (URLs) cited in $\mathcal{V}(s_i)$ and assess whether they support the claim under a strict support criterion (Appendix F).

| Model | Request. FulFill. | Analyt. Sound. | Struct. Cohere. | Format & Style | Inform Int. | Inform. Suff. | Ethics | Mean |
|---|---|---|---|---|---|---|---|---|
| **General LLMs** | | | | | | | | |
| Qwen3-235B (fast) | 4.66 | 5.20 | 6.39 | 7.54 | 1.23 | 4.20 | 7.03 | 5.18 |
| Gemini 2.5 Flash (fast) | 4.81 | 5.46 | 6.61 | 7.71 | 1.35 | 3.99 | 7.39 | 5.33 |
| Claude Opus 4.5 (fast) | 4.94 | 5.44 | 6.77 | 7.90 | 2.33 | 4.50 | 7.60 | 5.64 |
| GPT-5.2 (fast) | 4.30 | 4.85 | 6.38 | 7.22 | 1.07 | 3.13 | 7.20 | 4.88 |
| **LLMs+Reasoning** | | | | | | | | |
| Qwen3-235B (think) | 5.03 | 5.32 | 6.65 | 7.92 | 1.14 | 3.90 | 7.31 | 5.32 |
| Gemini 2.5 Pro (think) | 4.96 | 5.74 | 7.11 | **8.11** | 2.30 | 4.40 | 7.53 | 5.74 |
| Claude Opus 4.5 (think) | 5.08 | 5.70 | 6.89 | 7.92 | 2.27 | 4.22 | 7.63 | 5.67 |
| GPT-5.2 (think) | **5.54** | **6.21** | **7.20** | 8.02 | 2.11 | 4.16 | **7.97** | 5.89 |
| **LLMs+Reasoning+WebSearch** | | | | | | | | |
| Qwen3-235B (think+search) | 4.22 | 4.44 | 5.76 | 6.65 | 5.10 | 5.45 | 6.85 | 5.50 |
| Claude Opus 4.5 (think+search) | 4.57 | 5.14 | 6.15 | 7.18 | 6.92 | 7.62 | 7.30 | 6.41 |
| GPT-5.2 (think+search) | 5.50 | 6.05 | 7.13 | 7.94 | 5.57 | 6.17 | 7.95 | **6.62** |
| **Deep Research** | | | | | | | | |
| WebThinker (Li et al., 2025b) | 4.38 | 4.74 | 5.87 | 7.32 | 6.09 | 6.38 | 7.05 | 5.98 |
| Qwen3-235B (deep) | 4.22 | 4.80 | 5.12 | 7.05 | 6.44 | **7.90** | 7.12 | 6.09 |
| Gemini 2.5 Pro (deep) | 4.66 | 5.56 | 6.53 | 7.41 | 5.90 | 7.61 | 7.25 | 6.42 |
| Claude Opus 4.5 (deep) | 4.51 | 5.31 | 5.89 | 7.17 | 5.96 | 5.66 | 7.37 | 5.98 |
| OpenAI (deep) | 4.79 | 5.33 | 6.49 | 7.51 | **6.97** | 6.89 | 7.52 | 6.50 |

*Table 2.* Evaluation results for expert reports generated by baseline methods. The best score in each column is shown in **bold**, and the second highest score is underlined.

We then summarize claim-level outcomes into fine-grained metrics aligned with the Integrity/Sufficiency subdimensions (*e.g.*, Claim Factuality, Citation Support, Evidence Coverage, and Reference Reliability/Diversity), and aggregate these metrics into the corresponding final dimension-level scores (Appendix F.4). Implementation details for long-document processing and efficiency-oriented engineering (*e.g.*, batching and grouping) are provided in Appendix F.4.

## 5. Experiment Setup

### 5.1. Implementation Details

We evaluate report quality using an LLM-as-a-judge approach across five dimensions—Request Fulfillment, Analytical Soundness, Structural Coherence, Format & Style, and Ethics Compliance. The LLM judge receives the task query, the report being evaluated, task-specific Expert Evaluation Guidance, and a fixed set of fine-grained rubric items. It assigns a 1–10 score and a brief rationale to each item and returns the results as a JSON object mapping each item to its score and rationale. In parallel, Information Integrity and Information Sufficiency are assessed by the Information Verification Module, which extracts and type-classifies claims (Types A–F) and verifies the verifiable subset (Types A–C) against evidence documents retrieved from the report's cited sources (URLs), with semantic back-tracking to recover omitted citations for implicit claims. The module outputs quantitative metrics aligned with the Integrity/Sufficiency subdimensions (*e.g.*, Claim Factuality, Citation Support, Evidence Coverage, Reference Reliability/Diversity), and hierarchically aggregates them into the corresponding dimension-level scores (Appendix D.4). We report results using GPT-5.2 for Report Quality Assessment and GPT-5-mini for the Information Verification module, with an average evaluation cost of approximately 0.5–1.0 per report. We select these judge backbones based on backbone ablations that jointly consider validation performance and cost efficiency; see Appendix I.

### 5.2. Baseline Models

To compare expert report generation performance, we consider four baseline families: General LLMs (*fast*), LLMs+Reasoning (*think*), LLMs+Reasoning+WebSearch (*think+search*), and Deep Research (*deep*). Each family is instantiated using multiple model backbones—Qwen, Gemini, Claude, and GPT (Yang et al., 2025; Google, 2025; Anthropic, 2025; OpenAI, 2025b), and they differ in whether they (i) use reasoning, (ii) use web search, and (iii) employ research-system orchestration. Detailed model configurations are provided in Appendix G.

## 6. Experiments

### 6.1. Main Results

Summarizing Table 2, most models perform relatively well on structural coherence, format/style, and ethics, but lag on request fulfillment and analytical soundness. This suggests that presentation-oriented aspects of report writing are relatively mature, whereas expert-level intent alignment and reasoning completeness remain limited. Across baseline families, adding reasoning in *think* improves overall performance over *fast*. Moreover, external-information metrics—Information Integrity and Information Sufficiency—see fur-

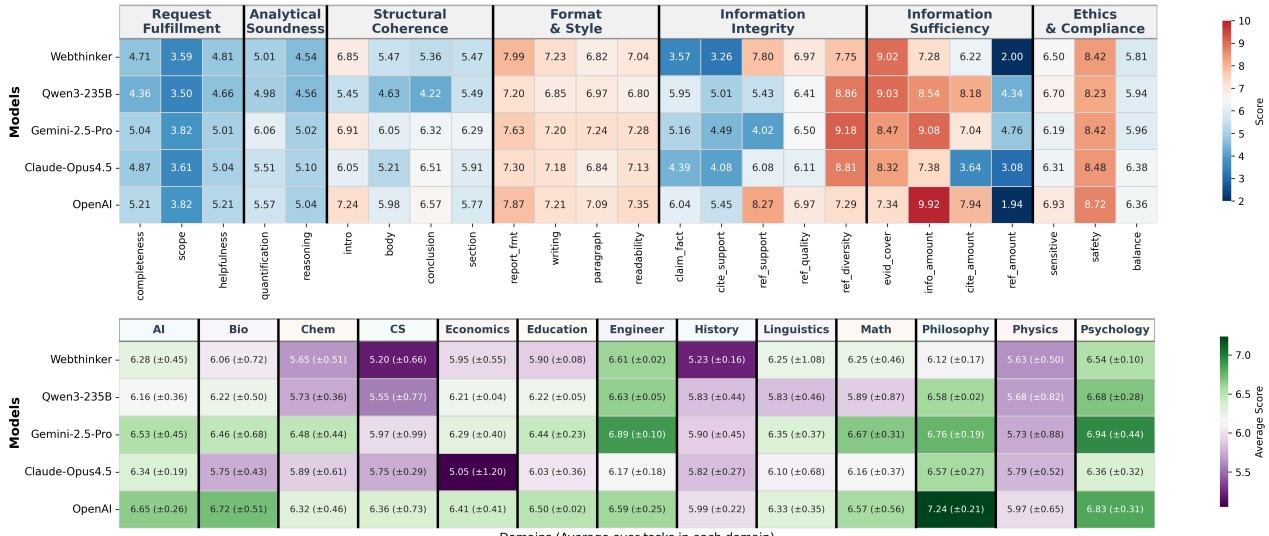

*Figure 3.* Heatmap visualizations of expert report evaluation results. (a) Evaluation scores by subdimension (b) Evaluation scores by domain, reported as mean (std).

ther gains with *think+search* and *deep*. An interesting finding is that reasoning models without web search (*think*) outperform *think+search* and *deep* on report-quality metrics excluding information-related scores. This suggests that integrating diverse external information can blur the problem definition and argument structure. Accordingly, simply adding search and external sources may not guarantee improved report-writing quality.

### 6.2. Fine-grained Analysis

Figure 3(a) presents fine-grained report-evaluation results for Deep Research systems. Most subdimensions under *Request Fulfillment* and *Analytical Soundness* receive low scores. Among them, the *scope* score under *Request Fulfillment* is particularly low, indicating that reports often fail to clearly specify what they cover and do not cover, as well as the assumptions and constraints underlying these scope decisions. In addition, *ref_amount* is low under *Information Sufficiency*, suggesting that systems tend to rely on a small number of references rather than leveraging a broad set of sources.

Figure 3(b) shows domain-level performance of Deep Research agents. We observe variation across disciplines: agents generally achieve higher scores in *AI*, *Engineering*, and *Psychology*, while scores are lower in *History* and *Physics*. Taken together, these results demonstrate that our evaluation framework offers a more detailed view of agent performance than a single aggregate score. By showing which evaluation dimensions and domains agents struggle with, the framework enables a more precise diagnosis of their limitations and provides clearer guidance for future improvement.

| Evaluation Method | Pearson $r$ | Spearman $\rho$ | Pairwise Agr. |
|---|---|---|---|
| Vanilla | 0.63(0.16) | 0.59(0.17) | 0.68(0.09) |
| + Dimensions | 0.66(0.11) | 0.62(0.14) | 0.77(0.07) |
| + Granular Rubrics | 0.64(0.12) | 0.59(0.08) | 0.76(0.04) |
| + Expert Guidance | 0.73(0.06) | 0.71(0.06) | 0.84(0.03) |
| Inter-Human | 0.81 | 0.74 | 0.79 |

*Table 3.* Average correlation with human evaluations across five LLM-based judges under incremental addition of evaluation components (reported as mean(std)). Inter-Human shows inter-annotator agreement.

### 6.3. Correlation with Human Evaluation

To validate the proposed evaluation method and assess the contribution of each component, we compared LLM-based evaluations against human expert judgments. For each of the 45 reports, we collected two independent ratings from domain experts whose expertise aligned with the report's topic (90 ratings total). We computed Pearson's r, Spearman's ρ, and LLM–human pairwise agreement for each of the five LLM-based evaluator models, and report the average of each metric across models. Additional details on the experimental setup and human evaluation protocol are provided in Appendix H.

Table 3 shows how alignment with human judgments changes as evaluation components are added step by step. Vanilla, which assesses overall quality holistically, and +Dimensions, which introduces high-level evaluation dimensions, both achieve relatively high correlation with human judgments. However, Pearson's r and Spearman's ρ drop at the +Granular Rubrics stage where the rubric is further decomposed into many fine-grained items. In contrast, task-specific +Expert Guidance helps evaluators apply these

| Method | Krip. $\alpha$ | ICC(2,1) | ICC(2,k) |
|---|---|---|---|
| Vanilla | 0.46 | 0.48 | 0.82 |
| + Dimensions | 0.32 | 0.37 | 0.75 |
| + Granular Rubrics | 0.33 | 0.38 | 0.76 |
| + Expert Guidance | 0.55 | 0.56 | 0.87 |

*Table 4.* Inter-evaluator reliability across five LLM-based evaluation models. Krip. $\alpha$: Krippendorff's alpha. Higher values indicate more consistent evaluations.

| Model | Batch | Density | Recall | Cls. F1 | Bin. F1 | Cost ($) |
|---|---|---|---|---|---|---|
| Annotated | - | 7.22 | - | - | - | - |
| GPT-5.2 | 10 | 7.91 | **96.29** | 59.33 | 80.4 | 1.66 |
| | 20 | 7.06 | 92.72 | 62.60 | **80.8** | 0.33 |
| GPT-5-mini | 10 | 5.54 | 92.17 | **68.80** | 80.4 | 0.16 |
| | 20 | 5.19 | 90.66 | 67.52 | 79.0 | **0.10** |

*Table 5.* Claim extraction and classification results across models (low effort) and batch sizes. **Cls. F1** denotes 6-class F1 (Types A–F), and **Bin. F1** the binary verifiable (A–C) vs. non-verifiable (D–F) F1.

| Grouped | Retrieval | F1 | Cost ($/1k) |
|---|---|---|---|
| ✗ | ✗ | 87.25 | 12.84 |
| 10 | ✗ | 90.91 | 3.65 |
| 10 | ✔ | 83.10 | **0.95** |
| 20 | ✗ | **91.61** | 3.46 |
| 20 | ✔ | 87.25 | **0.95** |

*Table 6.* Ablation study on GPT-5-mini (low effort) comparing grouped verification and retrieval.

rubric items by surfacing domain-relevant cues that non-experts may overlook, thereby achieving the highest correlation with human judgments. Complementing this step-wise ablation, we also conduct a leave-one-component-out analysis for Report Quality Assessment to further examine the contribution of each component. The results are reported in Appendix J.

### 6.4. Inter-evaluator Reliability

To verify that the proposed evaluation method yields consistent results across different LLM evaluators, we measure inter-evaluator reliability (Artstein, 2017; Lee et al., 2025). Specifically, we compute Krippendorff's $\alpha$ and the intraclass correlation coefficient (ICC) using the scores from the five LLM judges described in Section 6.3. As shown in Table 4, Vanilla achieves moderate inter-evaluator reliability, but reliability drops substantially under +Dimensions. +Granular Rubrics yields a slight recovery, though still below Vanilla. +Expert Guidance attains the highest reliability across metrics. This suggests that expert guidance clarifies what to look for under each criterion, enabling more consistent judgments across different LLM judges.

### 6.5. Information Verification Module Evaluation

To evaluate the proposed Information Verification module, we validate each stage of the pipeline against expert-annotated ground truth. The human-annotation setup for claim extraction, classification, and semantic back-tracking is detailed in Appendix E.2, and back-tracking performance is reported in Table 15. For claim verification, the annotation protocol and LLM procedure are detailed in Appendix E.3 and F.3. Table 19 reports the llm judge selection for the module, and Table 22 reports its robustness to the choice of LLM evaluator.

**Claim Extraction Analysis**   Table 5 demonstrates claim extraction and claim classification performance. Claim extraction performance is measured by *Density*, which captures whether the system extracts a sufficient number of independent claims per paragraph, and *Recall*, which measures the share of ground-truth claims semantically covered by the extracted claims as judged by an LLM. Claim classification performance is further evaluated by *Cls. F1*, the

6-class F1 over claim types (A–F), and *Bin. F1*, the binary F1 for distinguishing verifiable claims (A–C) from non-verifiable claims (D–F), which determines whether a claim is passed to the downstream verification stage. *Batch* denotes the number of report sentences input per call; a larger batch size lowers cost.

*GPT-5.2* achieves the highest *Recall* (96.29%) at batch size 10, but at substantially higher cost. In contrast, *GPT-5-mini* achieves a comparable *Recall* (92.17%) and the highest *Cls. F1* (68.80) at roughly an order of magnitude lower cost. Increasing the batch size causes only minor degradation in *Recall*, *Cls. F1*, and *Bin. F1*, while reducing cost to about two-thirds. We therefore adopt the batch-size-20 configuration of *GPT-5-mini* for large-scale claim extraction, balancing cost and performance.

**Claim Verification Performance**   Table 6 reports the performance of the claim verification module. Here, grouped verification jointly verifies claims associated with the same evidence URL, and retrieval uses only claim-relevant evidence chunks instead of the full evidence context. The results show that grouped verification substantially reduces cost while maintaining or improving F1, and retrieval further reduces cost with only a modest reduction in F1. We therefore adopt the group-size-20 configuration with retrieval as the final pipeline setting, balancing cost and performance.

## 7. Conclusion

We propose DEER, a benchmark and evaluation framework for systematically assessing deep research reports. DEER builds a hierarchical taxonomy, instantiates it as 101 fixed rubrics, and provides task-specific expert guidance to im-

prove the reliability of LLM-based judging. Beyond rubric scoring, DEER evaluates information use by tracing all claims back to their external information sources to verify evidence and quantifying evidence quality, enabling a more complete report-level assessment. Experiments show that deep research systems perform well on structure/style and information use, but remain limited in meeting expert-level requirements and producing analytically sound analyses. With its taxonomy, fixed rubrics, and quantitative metrics, DEER supports fine-grained diagnosis and systematic improvement beyond mere performance assessment.

## Limitations

While our evaluation framework relies on LLM-based judges, which may inherently exhibit biases relative to human experts, our extensive validation shows a high correlation with human judgment, suggesting that these biases are systematic and manageable. Furthermore, although our current benchmark focuses on text-based reports, this specialization enables a deeper, more rigorous analysis of information integrity and logical coherence, laying a solid foundation for future extensions to multimodal research tasks.

## Impact Statement

This paper presents DEER, a benchmark for the systematic and reliable evaluation of deep research agents on expert report generation. By assessing agents' ability to synthesize information, ground claims in evidence, and generate long-form reports for complex research tasks, DEER supports structured comparison of these systems and may inform downstream agent selection and deployment decisions.

As with any automated evaluation framework, inappropriate or uncritical use of DEER may lead to over-reliance on quantitative metrics without sufficient human judgment. We therefore emphasize that DEER is intended to complement, not replace, human oversight in the evaluation and use of deep research agents.

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

| Benchmark | Human rubrics | Expert curated | Open-ended | Non-tech domains | LLM judge | Claim fact-check | Interpret-ability | Avg. rubrics |
|---|---|---|---|---|---|---|---|---|
| AcademicBrowse (Zhou et al., 2025) | ✗ | ✗ | ✗ | ✓ | ✗ | ✗ | ✗ | – |
| BrowseComp (Wei et al., 2025a) | ✗ | ✗ | ✗ | ✓ | ✗ | ✗ | ✗ | – |
| Mind2Web2 (Gou et al., 2026) | ✗ | ✓ | ✗ | ✓ | ✓ | ✗ | △ | 50 |
| ExpertLongBench (Ruan et al., 2026) | ✓ | ✓ | ✓ | ✓ | ✓ | ✗ | ✗ | 16 |
| ResearchBench (Liu et al., 2025) | ✗ | ✗ | ✗ | ✓ | ✗ | ✗ | ✗ | – |
| ResearcherBench (Xu et al., 2025) | ✓ | ✓ | ✓ | ✗ | ✓ | △ | △ | 14 |
| ReportBench (Li et al., 2025a) | ✗ | ✗ | ✗ | ✓ | ✓ | ✓ | △ | – |
| DeepScholar-Bench (Patel et al., 2025) | ✗ | ✗ | ✗ | ✗ | ✓ | △ | △ | – |
| LiveDRBench (Java et al., 2026) | ✗ | ✗ | ✗ | ✓ | ✓ | △ | ✗ | – |
| SPOT (Son et al., 2025) | ✓ | ✗ | ✗ | ✗ | ✓ | △ | ✗ | – |
| DeepResearchGym (Coelho et al., 2025) | ✗ | ✗ | ✓ | ✓ | ✓ | ✗ | △ | – |
| DeepResearchBench (Du et al., 2026) | ✗ | ✓ | ✓ | ✗ | ✓ | △ | △ | 25 |
| DeepResearchArena (Wan et al., 2026) | ✗ | ✗ | ✓ | ✓ | ✓ | △ | △ | – |
| RigorousBench (Yao et al., 2025) | ✓ | ✓ | ✓ | ✓ | ✓ | ✗ | ✓ | 61(48+13) |
| LiveResearchBench (Wang et al., 2026) | ✗ | ✓ | ✓ | ✓ | ✓ | △ | △ | – |
| ResearchRubrics (Sharma et al., 2026) | ✓ | ✓ | ✓ | ✓ | ✓ | ✗ | △ | 26 |
| **DEER (Ours)** | ✓ | ✓ | ✓ | ✓ | ✓ | ✓ | ✓ | **101(+10)** |

*Table 7.* Comparison of DEER with representative deep research benchmarks. Here, ✓ indicates full support, ✗ no support, and △ partial support. For *Claim fact-check*, △ means that only a subset of claims (e.g., explicitly cited or gold-labeled ones) are checked rather than all explicit and implicit claims. For *Interpretability*, △ indicates that the benchmark offers only coarse, dimension-level insight (e.g., a few high-level scores), rather than a shared rubric-item–level diagnostic breakdown that is consistent across tasks. For DEER, (+10) means the number of information verification metrics.

## A. Comparison with Deep Research Benchmarks

In this section, we compare existing deep research benchmarks with our methodology. Table 7 extends and modifies the comparison table proposed in ResearchRubrics (Sharma et al., 2026), and adds new evaluation axes to summarize how DEER differs from prior work. The ResearchRubrics table uses five axes (whether rubrics are human-written, whether experts curate tasks, whether tasks are open-ended, whether non-technical domains are included, and whether an LLM-as-judge is used). On top of this, we add two axes that are essential for deep research report evaluation: (1) Claim fact-check and (2) Interpretability.

### A.1. Benchmark Landscape

**Search and browsing agent benchmarks.** The first block of the table consists of benchmarks for evaluating search and browsing agents. AcademicBrowse (Zhou et al., 2025), BrowseComp (Wei et al., 2025a), and Mind2Web2 (Gou et al., 2026) evaluate, respectively, the ability to browse an academic corpus or the open web to produce short answers to complex queries, the ability to perform agentic search across diverse websites, and the consistency between generated answers and cited sources. These benchmarks focus on "how well the system finds information," i.e., search and browsing strategies, and thus are one step removed from the *deep research report quality evaluation* that we study.

**Benchmarks for partial abilities or conceptualizations of deep research.** The second block covers benchmarks that focus on specific component abilities or conceptualizations of deep research. ExpertLongBench (Ruan et al., 2026) evaluates the ability to generate long-form expert text without external search. ResearchBench (Liu et al., 2025) evaluates the ability to extract inspirations from papers and generate research hypotheses. ResearcherBench (Xu et al., 2025) evaluates long-form responses to frontier AI research questions using a dual framework: expert-rubric insight quality and citation-based factuality (faithfulness/groundedness). ReportBench (Li et al., 2025a) and DeepScholar-Bench (Patel et al., 2025) assess academic survey/related-work reports, focusing primarily on literature selection and citation-grounded verifiability of report content. LiveDRBench (Java et al., 2026) evaluates the recovery of correct claims in search tasks that require many information units and non-trivial reasoning. SPOT (Son et al., 2025) measures how well a system can detect critical errors in published papers. These benchmarks finely evaluate partial abilities such as expert writing, hypothesis generation, claim discovery, and error

detection, but it is difficult to view them as evaluating the overall quality of reports produced by web- or literature-based deep research agents.

**Benchmarks for deep research report quality.** The third block targets benchmarks that evaluate the quality of deep research reports themselves. DeepResearchGym (Coelho et al., 2025) provides an offline web-corpus sandbox with an LLM-as-judge protocol. DeepResearchBench (Du et al., 2026) evaluates multi-domain web-based deep research reports using LLM-generated evaluation criteria (RACE), citation-based fact-checking (FACT), and dimensions including coverage, depth, presentation, and citation accuracy.

**Concurrent work.** Concurrently with DEER, several benchmarks have further advanced report-level evaluation. Deep-ResearchArena (Wan et al., 2026) derives tasks from seminar transcripts and evaluates evidence–keypoint alignment (KSR/KCR/KOR) with task-specific checklists (ACE). RigorousBench (Yao et al., 2025) uses expert-curated queries and two-level human rubrics (GRR/QSR), and adds a trustworthiness signal via matching citations to curated trustworthy-source links (TSL). LiveResearchBench (Wang et al., 2026) evaluates web-based deep research reports in a live, multi-domain setting using LLM-judged criteria, e.g., presentation & organization, coverage & comprehensiveness, and citation accuracy. ResearchRubrics (Sharma et al., 2026) provides 2,500+ expert-written rubric items spanning axes such as explicit requirements, synthesis, and reference use, with mandatory vs. optional criteria per task.

## A.2. Key Differences from Prior Work

**Alignment with expert standards.** A key concern in deep-research evaluation is whether the reported score truly reflects expert notions of report quality. In several benchmarks (Du et al., 2026; Wang et al., 2026; Wan et al., 2026), LLMs are used not only as judges but also to instantiate parts of the evaluation criteria (e.g., LLM-generated dimensions or task-specific checklists), which can leave ambiguity about how closely evaluation aligns with expert standards; moreover, even with well-defined criteria, LLM judges may apply them unreliably due to limited domain knowledge and weak evidence judgment. In contrast, DEER anchors evaluation in a shared rubric system grounded in established expert reporting norms and guidelines, and further provides task-specific expert guidance so that LLM-based scoring better aligns with expert judgment rather than ad hoc scoring heuristics.

**Claim-level fact checking.** Some benchmarks (Du et al., 2026; Wang et al., 2026) perform citation-based verification only for cited claims, leaving uncited claims unchecked. Another approach (Wan et al., 2026) scores alignment to keypoints extracted from cited URLs, making verification conditional on what is cited and less suited to detecting missing evidence. In contrast, DEER extracts all claims from a report and, for each claim, (i) determines whether evidence is required, (ii) links not only explicitly cited sources but also recovers omitted citation links by tracing each claim back to earlier cited context in the report, and (iii) verifies whether the linked evidence supports the claim. For more detailed comparisons, see Table 8.

**Systematic interpretability.** Beyond a single overall score, deep-research evaluation should support consistent diagnosis of failure modes across tasks. When benchmarks use task- or prompt-specific sub-criteria, fine-grained diagnostics are not standardized across tasks, so results tend to be interpretable mainly at coarse, high-level dimensions and are difficult to compare at a shared checklist level (Du et al., 2026; Wang et al., 2026; Wan et al., 2026; Sharma et al., 2026). DEER instead uses a hierarchical, shared rubric taxonomy grounded in established expert report-writing norms and guidelines, applying a fixed set of rubric sub-dimensions and items across tasks. As a result, DEER evaluates each report against a dense set of rubric items for more thorough, fine-grained assessment, and supports rubric-item-level diagnosis of system weaknesses.

# B. Data Construction Details

## B.1. Topic Domain Analysis

To construct deep research tasks, we analyzed 5,842 in-house user queries collected from our deep research system to estimate real-world domain demand and to guide the benchmark's target domain distribution. Figure 4 shows the resulting distribution. Based on this analysis, we finalized 11 topic domains: the 10 most frequent domains and an *Others* category that aggregates all remaining domains. We used only aggregated topic counts/statistics derived from in-house queries; no raw queries were released or included, and all queries complied with the organization's privacy policy, including the removal of any personally identifiable information.

| Benchmark | Explicit Verif. | Implicit Verif. | Global Context | Key Limitation vs. DEER |
|---|:---:|:---:|:---:|---|
| LiveDRBench (Java et al., 2026) | ✗ | ✗ | ✗ | Relies on matching against static *Ground Truth* claims; lacks dynamic verification of citations against web sources. |
| DeepResearch Bench (Du et al., 2026) | ✓ | ✗ | ✗ | Ignores uncited sentences; fails to detect hallucinations in transitional logic. |
| DeepResearchGym (Coelho et al., 2025) | ✓ | ✗ | ✗ | Limited to explicitly cited claims; only calculates citation precision/recall metrics. |
| ResearcherBench (Xu et al., 2025) | ✓ | ✗ | ✗ | Treats uncited claims simply as "ungrounded" (empty URL) without attempting context-based verification. |
| DeepResearch Arena (Wan et al., 2026) | ✓ | ✗ | ✗ | Evaluates *Source → Report* coverage (summarization) rather than *Report → Source* verification; misses uncited hallucinations. |
| LiveResearchBench (Wang et al., 2026) | ✓ | ✗ | ✗ | Checklist- and judge-based evaluation depends on task-specific annotations and LLM judgment; implicit claims and cross-sentence dependencies are not systematically enumerated or verified at the claim level. |
| SPOT (Son et al., 2025) | ✓ | △ | ✗ | Evaluates internal error detection against static ground truth; penalizes valid but unannotated error predictions (false positives). |
| ReportBench (Li et al., 2025a) | ✓ | ✓ | ✗ | Verifies non-cited claims via *external* web search voting, ignoring internal document grounding. |
| DeepScholar-Bench (Patel et al., 2025) | ✓ | ✓ | ✗ | Relies on physical distance (sliding window $w$); misses long-range semantic dependencies. |
| **DEER (Ours)** | ✓ | ✓ | ✓ | **Systematically resolves implicit dependencies via semantic back-tracking to verify claims against the report's evidence.** |

*Table 8.* Comparison of Information Verification Pipelines.

| In-house Domain | Humanity's Last Exam Subject |
|:---:|:---:|
| Finance & Business | Economics |
| Software Development* | Computer Science, Artificial Intelligence |
| Science & Technology | Mathematics, Physics, Chemistry |
| Industrial / Hardware | Engineering |
| Education & Jobs | Education |
| Health | Biology, Psychology |
| Others | History, Linguistics, Philosophy |

*Table 9.* Mapping between in-house topic domains and Humanity's Last Exam subjects. *Combines "Software Development" and "Software" categories.

## B.2. HLE Subject Mapping

Although our deep research service logs reflect research-oriented usage rather than general-purpose QA, most users are not domain experts, and their queries are typically not formulated as prompts for expert-level academic reports or papers. Moreover, the logs contain many context-dependent fragments (e.g., follow-up queries within an ongoing session) and pragmatic information needs. Therefore, using these queries directly as evaluation tasks would likely mismatch the high-difficulty, expert-level report-generation setting we target in both scope and format.

Accordingly, we used Humanity's Last Exam (HLE) (Phan et al., 2025) as a source of expert-written, high-difficulty seed items that align with expert-level report generation. To preserve the 11-topic domain composition derived from actual deep research logs, we mapped each domain to one of HLE's 13 subject domains and sampled HLE items from the corresponding subjects as domain-specific seeds. At this time, we filtered candidates via a preliminary performance evaluation, excluding items that were excessively difficult for LLM-based evaluation and retaining only those within an appropriate difficulty range. As a result, we selected a total of 50 seed items across HLE's 13 subject domains. The correspondence between deep research topic domains and HLE subject domains is summarized in Table 9.

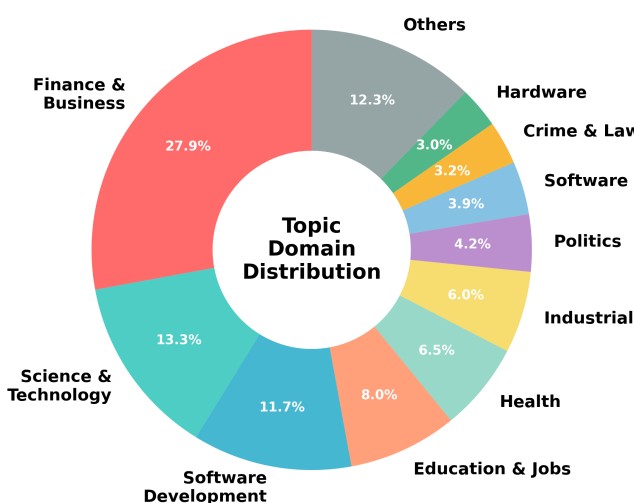

*Figure 4.* Topic domains extracted from real-world Deep Research service logs

### B.3. Conversion from HLE QA to Deep Research Reports

Because the selected HLE items are presented in a QA format, they are not directly suitable for deep research report-generation tasks in their original form. Accordingly, for each item, a domain expert reviewed the question, answer, and rationale to identify the underlying concepts, theories, and phenomena, and then reformulated it into a research-oriented task query appropriate for the deep research setting. During this reformulation, we removed answer-revealing elements from the prompt, such as specific answers, factual conclusions, and proofs, so that the model must derive the reasoning and conclusions on its own. When necessary, we also included writing guidance that constrains report development—such as the intended scope of analysis, comparative perspectives, and key issues to be addressed—to prevent uncontrolled drift and to enable more fine-grained evaluation of whether required elements are covered. Overall, this conversion reconstructs short-answer QA items into long-form report-generation tasks that require expert-level reasoning and exposition.

Each task query was drafted by one domain expert and cross-reviewed by another expert from the same field. Cross-review was repeated in multiple rounds as needed to check whether the reformulated task (i) is not overly narrow, (ii) requires expert-level domain expertise, (iii) specifies a sufficiently concrete research scope and direction, and (iv) does not contain excessive hints that could steer the model toward the answer or conclusion; the task was revised accordingly. Experts were individuals with a master's degree in the relevant field or equivalent domain expertise. An example research task is provided in Figure 5.

> **Example Research Task (MDP / Value Iteration)**
>
> Write a research report that explores the conditions under which value iteration in Markov Decision Processes converges geometrically to the optimal value function. Your report should provide precise definitions of the key concepts involved, examine the roles of rewards and discounting in ensuring convergence, and analyze both theoretical and practical factors that influence the algorithm's behavior. Discuss circumstances where convergence occurs, where it may fail, and highlight open questions or limitations. The report should be structured as a rigorous technical investigation, integrating mathematical reasoning, conceptual explanations, and illustrative examples to support the analysis.

*Figure 5.* Task Query Example

### B.4. Construction of Expert Evaluation Guidance

For each task, we constructed Expert Evaluation Guidance to specify what an expert report for the given prompt must cover. The Guidance includes only the mandatory elements required by the topic (excluding optional content or stylistic preferences) and describes each element in as concrete and verifiable a form as possible so that compliance can be judged.

| Dimensions | Sub-dimensions |
| --- | --- |
| Request Fulfillment | Completeness, Scope, Helpfulness |
| Analytical Soundness | Quantification, Reasoning |
| Structural Coherence | Introduction, Body, Conclusion, Section |
| Format & Style | Report Format, Writing Quality, Paragraph Quality, Readability |
| Ethics & Compliance | Sensitive Handling, Safety & Impact, Perspective Balance |
| Information Sufficiency | Evidence Coverage, Claim Amount, Citation Amount, Reference Amount |
| Information Integrity | Claim Factuality, Citation Support, Reference Support, Reference Quality, Reference Diversity |

*Table 10.* Deep Research Report Evaluation Taxonomy: 7 Major Dimensions and 25 Sub-dimensions.

The required elements are derived naturally from the task prompt and its writing requirements, and, when applicable, the Guidance also reflects key concepts implied by the underlying HLE item. In addition, because the prompt's writing requirements (writing-direction instructions/additional requested constraints) are intended to keep the report from deviating from the intended direction and to enable fine-grained evaluation, they are also included as mandatory elements in the Guidance.

For all 50 tasks, Expert Evaluation Guidance was written in the same expert workflow simultaneously with query reformulation, so that each task's requirements and evaluation criteria are mutually aligned. As in query reformulation, each Guidance was drafted by one domain expert and cross-reviewed by another expert from the same field. During cross-review, we checked and revised whether (i) the topic's mandatory content elements were included at a sufficiently detailed level and written in an evaluable way, (ii) optional content or stylistic preferences were not included as mandatory evaluation elements, and (iii) requirements specified in the task prompt's writing requirements (writing-direction instructions/additional requested constraints) were reflected without omission in the Guidance. An example Expert Evaluation Guidance is provided in Figure 6.

## C. Deep Research Report Evaluation Taxonomy Details

### C.1. Dimension and Criteria Specification

In this section, we detail the 7 dimensions and 25 sub-dimensions of the Deep Research Report Evaluation Taxonomy presented in the main text.

**Request Fulfillment**   Request Fulfillment evaluates whether the report meets the user's request at a professional standard. It is assessed along three sub-dimensions: Completeness, Scope, and Helpfulness. Completeness evaluates whether all elements explicitly stated or implicitly required by the query are addressed without omission and with sufficient depth. Scope evaluates whether, in addressing these elements, the report clearly specifies what is included and excluded, as well as its assumptions, constraints, and limitations, and maintains these consistently throughout. Helpfulness evaluates whether the report materially advances the user's goal by providing information that is sufficiently specific, practical, and actionable for direct use.

**Analytical Soundness**   Analytical Soundness evaluates how accurate and valid the report's figures and arguments are in terms of calculation, methodology, and logical development. Quantification examines whether calculation processes, used formulas/statistical models, and indicators/units are presented without error, are appropriate for the problem context, and are expressed transparently enough for a third party to reproduce and verify. Reasoning evaluates whether the argument is developed consistently with the topic, necessary background/assumptions/inference steps are specified, and major claims and counter-argument responses are persuasively supported without leaps based on facts, data, and interpretations.

**Structural Coherence**   Structural Coherence evaluates whether the introduction, body, and conclusion, and the structure of each section of the report, are organized consistently with the topic and scope. Introduction checks whether it concisely and clearly presents the topic, problem, significance, and basic scope. The Body checks whether the step-by-step argument is developed without omission or deviation, in accordance with the structure and scope presented in the introduction. Conclusion checks whether it synthesizes the body content to complete the purpose of the introduction without introducing new claims or evidence. Section checks whether each section supports the overall structure through clear organizational principles and appropriate connections.

**Format & Style**   Format & Style assesses whether the report faithfully follows the required format and style and is expressed in a way that the reader can read and understand the content without difficulty. Report Format checks whether external requirements, such as document length and section system, conform to professional report practices. Writing

Quality checks whether sentences are concise and accurate, and whether terminology and tone are consistent. Paragraph Quality checks whether paragraphs are sufficiently developed around a single point and naturally integrated with structural auxiliary elements. Readability assesses whether subheadings and simple explanations/examples are used to guide the reader in following complex content.

**Ethics & Compliance** Ethics & Compliance assesses whether the report is written ethically and responsibly regarding sensitive issues, potential harm, and perspective balance. Sensitive Handling checks whether sensitive topics such as politics, race, and gender are addressed with neutral, fair language and a balanced perspective. Safety & Impact checks whether negative impacts, side effects, and potential misuse of proposals/technologies/research results are adequately reviewed, and whether specific method presentations are dangerous. Perspective Balance assesses whether the discussion is balanced by appropriately including related perspectives and opposing views without bias toward a specific position.

**Information Sufficiency** Information Sufficiency assesses whether the information required to answer the request has been adequately secured and presented in terms of quantity and scope. Evidence Coverage evaluates whether verifiable evidence or reliable sources are provided to support claims that require external evidence. Claim Amount evaluates whether reliable facts and claims are sufficiently presented. Citation Amount evaluates whether sufficient citations are made at necessary points. Reference Amount is evaluated to determine whether the number of actually used references reaches an appropriate level.

**Information Integrity** Information Integrity assesses the factuality of the external information used in the report and the reliability and diversity of the citations/sources supporting it. Claim Factuality measures the proportion of verifiable claims that are determined to be factual. Citation Support measures the proportion of citations that actually support the claim among citations attached to each claim. Reference Support measures the proportion of sources that correctly support the argument among the presented/used references. Reference Quality assesses whether the sources used are reproducible and reliable. Reference Diversity assesses whether the evidence maintains a sufficient level of source diversity without becoming overly concentrated on a few documents.

### C.2. Construction Procedure and Evidence Mapping

To systematize and standardize universal and essential elements of expert report evaluation, this study synthesized and normalized evaluation standards across the natural sciences, engineering, and social sciences. To this end, we analyzed authoritative standards and guidelines widely used for writing and evaluating expert reports, such as guidelines for writing/reviewing academic papers and research reports, guidelines for systematic reviews/literature reviews, academic publication norms, and guidelines for writing/evaluating policy/regulatory/market analysis reports and consulting/advisory reports. We extracted the common required elements from these materials and integrated/generalized overlapping or domain-specific items.

To verify the validity of the drafted dimensions and sub-dimensions, this study conducted cross-validation with 10 independent experts representing 6 domains (Computer Science, Artificial Intelligence, Biology, Chemistry, Mathematics, Psychology). Two experts per domain reviewed the suitability of each item, focusing on factors considered necessary in light of domain practices, such as whether each dimension and criterion is actually a meaningful and necessary evaluation standard in that domain, whether the definition and scope of application are excessively ambiguous, and whether it unnecessarily overlaps with other criteria. Based on this, they made binary (pass/fail) evaluations for each criterion. The final evaluation framework included only the criteria that passed this cross-validation process.

The following presents the mapping of which external standards were referenced for each evaluation dimension.

**Request Fulfillment** Completeness aligns with guideline-level completeness checks that evaluate whether all required components of a report are present according to prescribed inclusion mandates across authoritative standards (Page et al., 2021; Schulz et al., 2010; International Organization for Standardization, 2011; What Works Clearinghouse, 2022; Percie du Sert et al., 2020; Global Reporting Initiative, 2023; von Elm et al., 2007). Scope reflects boundary-setting requirements that assess whether a report explicitly specifies its operative limits—such as eligibility conditions, assumptions, and constraints—within the structural fields defined in major evaluative frameworks (Page et al., 2021; National Institute of Standards and Technology, 2023; American Historical Association, 2024; American Economic Association, 2024; Center for Open Science, 2024; NASA, 2016; U.S. Securities and Exchange Commission, 2024; Chan et al., 2013). Helpfulness corresponds to coherence and utility standards that evaluate whether the report's core conclusions not only align logically with the evidence base but also deliver sufficient specificity and practical feasibility to comprehensively address the user's inquiry (Guyatt et al., 2013; American Psychological Association, 2025; Chang et al., 2024; Institute of Education Sciences, 2022; OECD, 2020).

**Analytical Soundness**    Quantification reflects evaluative requirements that check whether quantitative statements in a report correspond to verifiable computations, prespecified analytic procedures, and reproducible numerical evidence as established in major methodological frameworks(Guyatt et al., 2013; Joint Committee for Guides in Metrology, 2008; Association for Computing Machinery, 2025; International Union of Pure and Applied Chemistry, 2007; International Union of Pure and Applied Physics, 2010; The Econometric Society, 2024; American Educational Research Association et al., 2014; Mohr et al., 2024; OECD, 2011; International Accounting Standards Board, 2024; IPCC, 2019); see also ISO 30414:2018[3]. Reasoning aligns with reasoning-assessment criteria that evaluate whether inferential steps are explicitly grounded in evidence, free of unsupported leaps, and consistent with theoretical or mathematical rigor (European Mathematical Society, 2025; American Philosophical Association, 2024; American Educational Research Association, 2006; American Mathematical Society, 2022; U.S. Congress, 2024; Tong et al., 2007).

**Structural Coherence**    Introduction corresponds to structural-orientation requirements that assess whether a report's opening section presents its scope and analytic pathway in accordance with prescribed organizational fields in established reporting and specification frameworks (Page et al., 2021; Schulz et al., 2010; IEEE Computer Society, 2018; IFRS Foundation, 2021; Gagnier et al., 2013). Body aligns with structural-progression criteria that evaluate whether the main sections develop the promised analytical sequence without omission or drift relative to the ordered components defined in recognized guideline structures (Page et al., 2021; Schulz et al., 2010; IEEE Computer Society, 2018; Eaton et al., 2024; Gagnier et al., 2013). Conclusion reflects closure-consistency requirements that examine whether final statements integrate preceding evidence without introducing unsupported expansions, consistent with the conclusion-governance rules embedded in major evaluative standards (Page et al., 2021; Schulz et al., 2010; Gagnier et al., 2013). Section-Level corresponds to intra- and inter-section coherence checks that assess alignment, ordering, and non-duplication of content in accordance with structured specification and reporting templates in authoritative frameworks (Page et al., 2021; IEEE Computer Society, 2018; IFRS Foundation, 2021).

**Format & Style**    Report Format corresponds to format-governance requirements that evaluate whether a report adheres to prescribed structural conventions for scientific communication as codified in established editorial and reporting standards (Page et al., 2021; Schulz et al., 2010; International Committee of Medical Journal Editors, 2025; Comrie, B. and Haspelmath, M. and Bickel, B., 2015; Society for Industrial and Applied Mathematics, 2024; International Phonetic Association, 1999; American Political Science Association, 2018; American Sociological Association, 2022). Writing Quality aligns with language-precision criteria that assess clarity, specificity, and terminological consistency according to recognized guidelines for accurate and unbiased scholarly expression (Schulz et al., 2010; International Committee of Medical Journal Editors, 2025; University of Chicago Press, 2024; Garner, 2019; 2013). Paragraph Quality reflects cohesion-assessment rules that examine whether individual paragraphs follow a coherent internal logic—anchoring topic statements, supporting evidence, and transitional structure—in line with authoritative reporting and specification frameworks (Page et al., 2021; IEEE Computer Society, 2018; Columbia Law Review et al., 2020; American Sociological Association, 2022). Readability corresponds to accessibility-oriented requirements that evaluate whether the narrative facilitates comprehension through appropriate signaling, explanatory devices, and presentation practices as outlined in major editorial and reporting guidelines (Page et al., 2021; International Committee of Medical Journal Editors, 2025; Garner, 2013; Global Reporting Initiative, 2023).

**Ethics & Compliance**    Sensitive Handling corresponds to ethical-screening requirements that evaluate whether sensitive domains are addressed with neutrality, respect, and responsible language in accordance with established editorial and reporting ethics standards (International Committee of Medical Journal Editors, 2025; Page et al., 2021; Oral History Association, 2024; Organization of American Historians, 2018; British Psychological Society, 2021; National Commission for the Protection of Human Subjects of Biomedical and Behavioral Research, 1979; American Political Science Association, 2022). Safety & Impact aligns with harm-assessment provisions that examine whether potential adverse effects, misuse risks, or disproportionate impacts are identified and mitigated under recognized guidelines for responsible scientific communication (International Committee of Medical Journal Editors, 2025; Guyatt et al., 2013; National Research Council, 2011; World Medical Association, 2013; Ecological Society of America, 2021). Perspective Balance reflects fairness-evaluation criteria that assess whether multiple viewpoints and counterpositions are represented without undue bias following principles of impartiality articulated in authoritative publication and reporting frameworks (International Committee of Medical Journal Editors, 2025; Page et al., 2021; British Philosophical Association, 2024; Australasian Association of Philosophy, 2023; CFA Institute, 2024; American Bar Association, 2023; American Association for Public Opinion Research, 2021).

---

[3]https://www.iso.org/standard/69338.html

**Information Sufficiency** Evidence Coverage aligns with evidence-provision requirements that evaluate whether claims are accompanied by verifiable supporting sources in accordance with established evidentiary and reporting guidelines (Page et al., 2021; Data Citation Synthesis Group, 2014; Chang et al., 2024; International Society for Stem Cell Research, 2025; CFA Institute, 2020). Claim Amount corresponds to content-adequacy criteria that assess whether a report supplies all necessary factual and contextual material needed to justify its reasoning under recognized completeness and transparency standards (Page et al., 2021; International Organization for Standardization, 2011; EQUATOR Network, 2025; Linguistic Society of America, 2024; The Journal of Organic Chemistry, 2025; U.S. Geological Survey, 2024; Global Reporting Initiative, 2023). Citation Amount align with source-attribution rules that examine whether in-text citations are provided at appropriate argumentative locations following authoritative norms for evidentiary traceability (International Committee of Medical Journal Editors, 2025; Wilkinson et al., 2016; IEEE, 2025; American Chemical Society, 2021; Columbia Law Review et al., 2020). Reference Amount are evaluated to determine whether the number of actually used references is appropriate and whether they collectively form a well-documented, traceable, and accessible evidence base (Stanford HAI, 2025; Data Citation Synthesis Group, 2014); see also the DA-RT principles[4].

**Information Integrity** Claim Factuality reflects evidence-verification provisions that assess whether factual assertions in a report correspond to verifiable sources and documented evidence traces within established evaluative frameworks (Wilkinson et al., 2016; American Physical Society, 2023; National Institute of Standards and Technology, 2023; Navas et al., 2024; National Institute of Standards and Technology, 2024; U.S. Securities and Exchange Commission, 2024). Citation Support corresponds to source-justification checks that evaluate whether cited references substantively support the claims they accompany according to recognized standards for evidential accountability (Wilkinson et al., 2016; American Sociological Association, 2018; American Historical Association, 2024; Berez-Kroeker et al., 2018). Reference Support aligns with source-quality and provenance requirements that examine whether referenced materials reliably and transparently support the arguments for which they are cited, in line with authoritative data-governance and reporting criteria (Wilkinson et al., 2016; Chang et al., 2024; Data Citation Synthesis Group, 2014; American Psychological Association, 2025; International Accounting Standards Board, 2024). Reference Quality evaluates whether the set of sources consists of reproducible, trustworthy, and methodologically sound information sources, ensuring that the report's evidentiary foundation is grounded in reliable materials (American Chemical Society, 2021; IEEE, 2025; Stanford Encyclopedia of Philosophy, 2025; World Medical Association, 2013; National Commission for the Protection of Human Subjects of Biomedical and Behavioral Research, 1979). Reference Diversity evaluates whether the evidence base avoids excessive concentration on a small number of sources and instead maintains a balanced level of source diversity, reducing the risk of narrow, biased, or selectively framed arguments (EQUATOR Network, 2025; Linguistic Society of America, 2024; American Association for Public Opinion Research, 2021).

Overall, this framework comprehensively integrates authoritative standards and guidelines from a wide spectrum of disciplines, ranging from quantitative sciences and engineering to the humanities, social sciences, and professional fields such as law and finance (see Table 11). By synthesizing the core principles shared across these diverse domains, this taxonomy prioritizes criteria that are universally recognized as essential for high-quality intellectual work. Consequently, the resulting evaluation model ensures that the generated reports not only adhere to domain-specific best practices but also satisfy the fundamental requirements of validity, clarity, and professional integrity demanded by the global research and practitioner communities.

## D. Rubric Structure

In this appendix, we summarize the rubric taxonomy and scoring procedure used to evaluate Deep Research reports.

### D.1. Hierarchical Levels

We describe the rubric in two parts: (i) a shared taxonomy of dimensions and sub-dimensions, and (ii) the scoring instantiation for report-quality assessment, which specifies criteria and rubric items.

**Level 1: Evaluation Dimensions (7)** The 7 upper dimensions presented in Table 10 represent different evaluation perspectives of report quality.

**Level 2: Sub-dimensions (25)** Each dimension is decomposed into finer sub-dimensions (2–5 per dimension), specifying which aspects to examine within that dimension.

---

[4]https://www.dartstatement.org/

### D.2. Criteria and Rubric Items

For the report-quality dimensions, we further specify criteria and rubric items. For the information-verification dimensions, we instead use metric-based measurements at the sub-dimension level; see Appendix F.4.

**Level 3: Criteria (46)** Criteria operationalize each sub-dimension into evaluation requirements that are directly assessable from the report. One or more criteria are defined under each sub-dimension, and each criterion corresponds to a concrete aspect of report content, such as "Inclusion of requested items" and "Specification of scope/limitations."

**Level 4: Rubric Items (101)** Rubric items at the fourth level decompose each criterion (Level 3) into atomic scoring units under two aspects, *Coverage (C)* and *Quality (Q)*. Each rubric item is labeled as either (Coverage) or (Quality), and rubric items serve as the minimum unit for scoring. Table 12 shows example criteria and their associated rubric items.

### D.3. Coverage and Quality

At Level 4, rubric items evaluate each criterion (Level 3) from two perspectives: *Coverage (C)* and *Quality (Q)*. Coverage assesses whether the criterion is covered without omission as required throughout the document, including requirements distributed across multiple locations or composed of multiple detailed components. In contrast, Quality assesses whether the written content exhibits sufficient depth, logic, and rigor under professional report standards, conditioning on what is actually written (i.e., "how well it is written"). By separating Coverage and Quality, we can independently measure (1) what is missing and (2) the quality of what is included in a long report. In our rubric, Level 4 consists of 101 rubric items in total, comprising 66 Coverage items and 35 Quality items. Both use a 1–10 point scale, and the interpretation of score bands is summarized in Table 13.

### D.4. Scoring and Aggregation

The score aggregation flow in this evaluation framework consists of *rubric item → criterion → sub-dimension → dimension*. First, rubric items (the lowest-level units) are evaluated with a 1–10 score (or N/A). Coverage (C) and Quality (Q) are computed separately at the rubric-item level, then integrated at the criterion level, and the resulting criterion scores are aggregated to the sub-dimension and dimension levels. N/A items are excluded from all average calculations.

Let $r$ be the report, and type $T \in \{C, Q\}$ represent Coverage and Quality, respectively. Let the score of each rubric item $i$ be $s_{r,i} \in \{1, \ldots, 10\}$ (or N/A), and let the set of non-N/A rubric items of type $T$ belonging to criterion $c$ be $I_{c,r}^T$. Let $\mathcal{C}_{s,r}$ be the set of criteria belonging to sub-dimension $s$, and let $\mathcal{S}_{d,r}$ be the set of sub-dimensions belonging to dimension $d$. Then, the rubric item → criterion → sub-dimension → dimension aggregation is defined as follows:

The Coverage/Quality score (criterion level) of report $r$ for criterion $c$ is

$$S_r^T(c) = \frac{1}{|I_{c,r}^T|} \sum_{i \in I_{c,r}^T} s_{r,i}, \quad T \in \{C, Q\}, \tag{2}$$

and if $I_{c,r}^T = \varnothing$, $S_r^T(c)$ is set to N/A.

The integrated criterion score $S_r(c)$ is set as the average of defined values among C/Q. Let the set of defined types for criterion $c$ be $\mathcal{T}_c = \{T \in \{C, Q\} \mid S_r^T(c) \neq \text{N/A}\}$.

$$S_r(c) = \frac{1}{|\mathcal{T}_c|} \sum_{T \in \mathcal{T}_c} S_r^T(c). \tag{3}$$

If $\mathcal{T}_c = \varnothing$, $S_r(c)$ is set to N/A.

The score of sub-dimension $s$ is defined as the average of criterion scores belonging to that sub-dimension.

$$S_r(s) = \frac{1}{|\mathcal{C}_{s,r}|} \sum_{c \in \mathcal{C}_{s,r}} S_r(c). \tag{4}$$

If $\mathcal{C}_{s,r} = \varnothing$, $S_r(s)$ is set to N/A.

The score of dimension $d$ is defined as the average of sub-dimension scores belonging to that dimension.

$$S_r(d) = \frac{1}{|\mathcal{S}_{d,r}|} \sum_{s \in \mathcal{S}_{d,r}} S_r(s). \tag{5}$$

If $\mathcal{S}_{d,r} = \varnothing$, $S_r(d)$ is set to N/A.

In summary, we compute $S_r^C(c), S_r^Q(c)$ by averaging rubric items within each criterion, integrate them to obtain the criterion score $S_r(c)$, then average criterion scores to obtain the sub-dimension score $S_r(s)$, and finally average sub-dimension scores to obtain the dimension score $S_r(d)$.

# E. Human-based Information Verification Protocol

## E.1. Overview

DEER's Information Verification Protocol is based on a stepwise information verification procedure performed by human evaluators. This section details the actual two-step procedure (Claim Extraction, Factual Accuracy Evaluation) performed by human evaluators.

**Design rationale.**   Our typology is based on the premise that a claim's verifiability is determined not by the presence of a citation marker but by two underlying properties: whether the claim requires external evidence, and where its supporting evidence resides. Structural recaps of the report's own content, common knowledge, and internally derived statements require no external evidence, whereas factual assertions about the world do. For claims that require evidence, the supporting source may appear as an inline citation, in another sentence or section of the report (for example, a citation shared at the end of a paragraph), or be absent entirely. Combining these two properties yields the six claim types (A–F): the first distinguishes claims that require evidence (A–C and F) from those that do not (D, E), while the second characterizes how the evidence is located, including the implicit cases (B, C) that motivate semantic back-tracking. The two-step procedure below—claim extraction and classification, followed by claim verification—operationalizes this typology and produces the expert ground truth against which the LLM-based implementation (Appendix F) is evaluated.

## E.2. Step 1: Claim Extraction and Classification

**Human procedure.**   Evaluators segmented the report into paragraphs and sentences (in 'Lx.Sy' format, where L denotes the paragraph index and S denotes the sentence index), reviewed each sentence individually to identify sentences containing claims, and extracted only the core claims. Pronouns were replaced with explicit references according to the context, and if a single sentence contained multiple claims, they were separated. All claims were classified into 6 types (A–F): Explicit Citation (A), Implicit – Same Section (B), Implicit – Previous Section (C), Structural Recap (D), No Citation Required (E), and Unknown Source (F). For A–C type claims, the corresponding citation or evidence position was recorded together. This process was performed on 2 randomly selected reports (total 728 claims), thereby constructing a Ground Truth for evaluating the recall of the extraction model.

Table 14 shows the definitions of the 6 claim types, and Table 1 shows examples of extracted sentences and classification results.

## E.3. Step 2: Claim Verification

**Human procedure.**   For 100 claims randomly selected from types A–C, two human evaluators independently assessed their factuality. The evaluation followed a single-criterion protocol determining whether the cited source explicitly supports (*Supported*) or lacks information/is irrelevant to (*Not Supported*) the content of the claim.

**Annotator Qualifications & Adjudication.**   The evaluators consisted of 2 individuals holding a master's degree or equivalent experience in the report's domain. The two evaluators made judgments independently, and for items where disagreement occurred, the final label unanimously agreed upon through discussion was established as the Ground Truth. Through this process, personal bias was excluded, and the objectivity of the evaluation was secured. The human verification results for these 100 claims were used as Ground Truth for the model performance evaluation in Section 6.5.

**Detailed Verification Rubric.**   Fact verification was strictly performed according to the following sub-dimensions:

- **Supported**: When the cited document clearly and directly includes the core facts (figures, causality, definitions, etc.) of the claim. The implied meaning in context must match the intent of the claim.

- **Not Supported**: When the basis for the claim cannot be found in the cited document, or the document is irrelevant to the topic.

- **Error**: When the verification process fails due to accessibility issues (e.g., HTTP 4xx/5xx errors, Paywall, Captcha) or processing errors, preventing content verification.

**Source Reliability Check**   Independently of the content verification, we also evaluate the trustworthiness of the source domain itself.

- **Reliable**: Trustworthy sources such as academic journals, official statistics, and authoritative institutions.

- **Unreliable**: Sources with low credibility, such as personal blogs, social media posts, or unverified community forums.

**Inter-human Agreement.** To verify the reliability of this protocol, we measured the agreement (Cohen's Kappa) between the two evaluators. The analysis result showed that a high level of agreement (Substantial Agreement) of $\kappa = 0.80$ was achieved in the **Claim Support** judgment. This suggests that the proposed verification criteria are objective and reproducible.

# F. LLM-based Information Verification Implementation

## F.1. Overview

The Information Verification Module is designed to automate the human verification protocol described above. The LLM analyzes the report sentence by sentence to extract and classify claims, and if necessary, retrieves external documents to verify their factuality. In this process, algorithms such as Batch Extraction, Back-tracking, and Relevant Context Filtering were applied to achieve both cost efficiency and accuracy.

## F.2. Claim Extraction and Classification

**LLM adaptation.** The Information Verification Module is designed to automate the human verification protocol described above. The LLM analyzes the report sentence by sentence to extract and classify claims, and, if necessary, retrieves external documents to verify their factual accuracy. In this process, algorithms such as Batch Extraction, Back-tracking, and Relevant Context Filtering were applied to achieve both cost efficiency and accuracy.

**Batch Extraction Strategy** To mitigate the "Lost-in-the-Middle (Liu et al., 2024)" phenomenon that occurs when processing long contexts and to increase cost efficiency, this study introduced a Batch Extraction strategy. After dividing the entire report $D$ into sentence units $S = \{s_1, s_2, \ldots, s_m\}$, they are processed in batches of a fixed size $B$ (in this study, $B = 20$). Each batch of processing provides the full report context ($D$) at the beginning of the prompt, but instructs the model to extract claims only for the sentences in the current batch ($s_i, \ldots, s_{i+B-1}$). Figure 7 shows the simplified prompt structure used for batch extraction, along with an example JSON output. This method induces the model to focus on local sentences while remaining aware of the entire context, achieving human-level claim-extraction performance.

**Claim Extraction Evaluation Setup.** To rigorously evaluate the claim extraction performance in Table 5, we employed an LLM-based Judge (GPT-5). Standard lexical metrics (*e.g.*, ROUGE (Lin, 2004), Exact Match) are unsuitable for this task because the generated claims may differ in wording or granularity (*e.g.*, one sentence split into multiple atomic claims) while preserving the same semantic meaning. We defined two key metrics: (1) Paragraph-level Semantic Recall: Measures whether each ground-truth claim is semantically covered by the extracted claims within the same source paragraph. The LLM Judge compares the ground truth claim against all candidate claims extracted from the same source sentence and determines if the core information is present. (2) Classification F1: Measures whether the LLM correctly classified the claim type for the extracted claims. The implementation code and LLM prompts used for the evaluation are included in the supplementary material.

**Binary Verifiability Classification.** While the 6-class classification (Types A–F) provides fine-grained type information, the operationally critical signal for the downstream verification stage is whether a claim requires external evidence at all. We therefore additionally report a binary metric that merges Types A–C (verifiable, requiring external evidence) versus Types D–F (non-verifiable: structural, internal, or unsupportable). Under this binary view, the same GPT-5-mini extraction (Batch 20) achieves an F1 of 79, indicating that claims requiring external evidence are reliably identified while non-verifiable claims are safely excluded from the verification stage.

**Semantic Back-tracking for Citation Recovery** The Information Verification Module uses a Backtracking algorithm to find evidence for claims without explicit citations (Types B and C). Type B (Same Section) and Type C (Previous Section) claims often share citations from previous sentences in context. The algorithm traces the 'evidence_position' (location ID of the reference target sentence, *e.g.*, L1.S3) recorded in the claim's metadata and adds the citations held by that target sentence to the citations of the current claim. This restores omitted citation relationships and allows the corresponding source to be reviewed together in the subsequent verification step.

Formally, for a set of claims $\mathcal{C} = \{c_1, c_2, \ldots, c_n\}$, each claim $c_i$ has a position $p_i$, type $t_i$, explicit citations $R_i$, and a reference position $ref_i$ (if $t_i \in \{B, C\}$). The Back-tracking function $f_{backtrack}(c_i)$ is defined as follows:

$$f_{\text{backtrack}}(c_i) = \begin{cases} R_j & \text{if } t_i \in \{B, C\} \text{ and } \exists c_j \text{ s.t. } p_j = ref_i \\ \emptyset & \text{otherwise} \end{cases}$$

Finally, the citation set used for verification becomes $R_i' = R_i \cup f_{backtrack}(c_i)$.

**Semantic Back-tracking Evaluation.** To validate the effectiveness of the LLM's targeted evidence prediction, we compared it with a "Sliding Window (Patel et al., 2025)" baseline on the subset of correctly classified B/C claims ($N = 131$). The sliding window method collects all citations within a window of size $k$ centered on the claim. As shown in Table 15, the proposed LLM method achieves the highest Jaccard Index (0.7070) and Precision (0.7109), outperforming the sliding window baselines ($k = 5, 10, 15$). While increasing the window size ($k$) improves Recall (up to 0.93), it significantly degrades Precision and Jaccard due to the inclusion of irrelevant citations. This result demonstrates that the LLM's "Evidence Position" prediction provides a precise pointer to the supporting evidence, which is crucial for efficient verification.

**Verification Coverage Analysis.** We additionally analyze how much of the verifiable content (Types A–C) is actually covered by the verification pipeline. In the evaluated reports, only 62.5% of all verifiable claims carry an explicit citation marker; the remaining 37.5% are implicit (Types B and C). Without semantic back-tracking, the verification module can therefore directly assess at most 62.5% of verifiable content. With back-tracking, citations are recovered for the implicit claims by inheriting from the dependent prior context. Even accounting for back-tracking prediction errors (Table 15), the effective verification coverage extends to approximately 91% of all verifiable claims.

### F.3. Claim Verification

**Context Retrieval** Using the entire report or long retrieved documents as input in the verification step is costly and can induce hallucinations due to unnecessary information (Noise). To solve this, we apply Context Retrieval. For the verification target claim $q$ and the retrieved document $D_{retrieved}$, the document is divided into chunks $K = \{k_1, k_2, \ldots, k_r\}$ (Size $\approx$ 1000 tokens). Then, the relevance score $Sim(q, k_j)$ between each chunk and the claim is calculated, and only the top $N$ (Top-K) chunks are selected and used as input for the verification model.

In this study, the BM25 (Robertson et al., 2009) and OpenAI's *text-embedding-3-large* (OpenAI) was used as the embedding model: The selected chunk set $K_{selected} = \{k \in K \mid rank(Sim(q, k)) \leq N\}$ is combined while maintaining the original document order to form the final context $C_{final}$.

$$C_{final} = \text{Concatenate}(\text{SortByPosition}(K_{selected}))$$

Through this process, token costs can be reduced by more than 80% while maintaining or improving verification accuracy.

**LLM Verification Logic.** The Information Verification Module's automatic verifier is prompt-engineered to determine whether the given context supports the claim, identical to the human protocol above. The LLM infers whether the claim's core content and numerical information match the source, then makes a final judgment.

**Augmented Dataset and Robustness Evaluation** While the human-annotated dataset provides high-quality ground truth, it exhibits significant class imbalance, with 82 "Supported" claims and only 5 "Not Supported" claims among 100 examples. This skew limits the ability to effectively evaluate the model's capacity to discern unsupported claims and mitigate hallucinations. To address this, we constructed an adversarial augmented dataset. This dataset was generated by systematically perturbing initially supported claims—specifically by negating semantic meanings or altering numerical values—to create plausible but factually incorrect statements (i.e., "Not Supported"). All augmented examples were rigorously reviewed by human evaluators to ensure they are strictly false or unsupported by the source text.

**Ablation Study on Retrieval Parameters** To identify the optimal configuration for the cost-efficient `gpt-5-mini` model, we conducted an ablation study varying Batch Size (10, 20), Reasoning Effort (Low, Medium, High), Retrieval Method (BM25 (Robertson et al., 2009), OpenAI's `text-embedding-3-large` (OpenAI)), and Context Size (Top-K=2, 4). Table 16 summarizes the results on both the Original and Adversarial (Augmented) datasets. We observed that increasing the retrieved context size from Top-K=2 to 4 improved accuracy on the Original dataset (*e.g.*, 77.0% → 79.3% for OpenAI, Low Effort), but increased the cost by approximately 35% ($0.95 → $1.28). The **Low Reasoning Effort** setting proved to be highly cost-effective, achieving comparable or superior performance to Medium/High effort while costing significantly less. Notably, the model demonstrated high robustness on the **Adversarial dataset**, maintaining high accuracy (>88%) across most configurations, suggesting that it effectively distinguishes unsupported claims even under perturbations. Consequently, prioritizing feasibility of large-scale verification (cost/throughput) while retaining strong robustness, we selected the configuration **Batch 20, Low Effort, Top-K=2, and OpenAI Embedding**. This setup offers a minimal cost of $0.95 per 1k claims while maintaining a strong accuracy of 77.0% (Original) and 88.5% (Adversarial), making it the most balanced choice for our resource-constrained high-volume verification pipeline.

**Example.**

> **Claim:** "Multi-junction solar cells achieve efficiencies above 45% in lab settings [1]."
> **Reference [1]:** Reports a 46.2% lab efficiency under concentrated light.

**Evaluation:** The reference explicitly states 46.2% efficiency, supporting the claim of "above 45%".
⇒ **Final Result: Supported**

## F.4. Evaluation Metrics

The evaluation metrics are designed to assess the Integrity and Sufficiency subdimensions by decomposing evidence use into complementary, claim-level signals. Rather than relying on a single aggregate score, we measure multiple failure modes of information use—factual incorrectness, unsupported attribution, unreliable or inaccessible sources, and insufficient evidence coverage—so that different weaknesses in evidence grounding can be diagnosed explicitly.

**Integrity Metrics**

- **Claim Factuality**: The proportion of claims verified as factual among claims requiring external evidence (Type A, B, C).

$$\text{Score} = \frac{|\text{Supported Claims (A, B, C)}|}{|\text{Total Verifiable Claims (A, B, C)}|}$$

- **Citation Support**: The proportion of citations that correctly support the corresponding claim among all citations.

$$\text{Score} = \frac{|\text{Supported Citations}|}{|\text{Total Citations}|}$$

- **Reference Support**: The proportion of references that actually contributed to content verification (Supported) among unique references shown in the report.

$$\text{Score} = \frac{|\text{Supported Unique References}|}{|\text{Total Unique References Shown}|}$$

- **Reference Reproducibility**: The proportion of references that were accessible during the actual verification process and for which the webpage in markdown format could be successfully retrieved using the Jina API (not Error).

$$\text{Score} = 1 - \frac{|\text{Error References}|}{|\text{Used References}|}$$

- **Reference Reliability**: The proportion of references that are both reliable sources and support the content among the used references.

$$\text{Score} = \frac{|\text{Reliable \& Supported References}|}{|\text{Used References}|}$$

- **Reference Diversity (Normalized HHI)**: Measures how evenly citations are distributed across used references using the Normalized Herfindahl-Hirschman Index (Rhoades, 1993). We define $s_i$ as the share of citations for reference $i$ among total citations ($s_i = c_i / \sum c$), and HHI as follows:

$$\text{HHI} = \sum_{i=1}^{N} s_i^2$$

Based on this, the Normalized HHI score (0–10) is calculated as:

$$\text{Score} = 10 \times \left(1 - \frac{\text{HHI} - 1/N}{1 - 1/N}\right)$$

**Sufficiency Metrics**

- **Evidence Coverage**: The proportion of claims verifiable with external evidence (Type A, B, C) among all claims.

$$\text{Score} = \frac{|\text{Claims (A, B, C)}|}{|\text{Total Claims}|}$$

- **Information Amount**: The total number of claims verified as factual (Supported).

$$\text{Score} = |\text{Accurate Verifiable Claims}|$$

- **Citation Amount**: The total number of valid citations (Supported Citation) supporting claims.

$$\text{Score} = |\text{Supported Citations}|$$

- **Reference Amount**: The total number of valid references (Supported Reference) supporting claims.

$$\text{Score} = |\text{Supported References}|$$

**Final Score Calculation**   The final scores for Integrity and Sufficiency are computed by hierarchically aggregating the metrics (Metric $\rightarrow$ Criterion $\rightarrow$ Dimension), consistent with the `score_avgs` and `criteria_avgs` structure in the output.
**1. Normalization (Metric Level)**
Each raw metric value is first converted to a 0–10 scale:

- **Ratio-based metrics** (e.g., Factuality): Scaled linearly.

$$\text{Score} = \min(\max(R, 0), 1) \times 10$$

- **Quantity-based metrics** (e.g., Counts): Scored via step function with divisors $D$ (Info=15, Cit=10, Ref=4).

$$\text{Score} = \min\left(\left\lfloor \frac{\max(N-1, 0)}{D} \right\rfloor + 1, 10\right)$$

**2. Aggregation**

- **Criterion Level**: Average of normalized metric scores within each criterion.

- **Dimension Level**: Average of criterion scores within each dimension.

## G. Baseline Model Details

We use the following backbone model families in our experiments: Qwen3-235B, Gemini 2.5, Claude Opus 4.5, and GPT-5.2. Since Gemini 2.5 Pro includes reasoning by default, we use *Gemini 2.5 Flash* for the *fast* (non-reasoning) setting, and *Gemini 2.5 Pro* for the other settings. For *think*, we use each service's default reasoning budget. For *think+search*, we do not build a custom retrieval pipeline; instead, we use the built-in web search system provided by each service. We exclude Gemini *think+search* because citation information is not provided in its outputs. For WebThinker (Li et al., 2025b), we use Qwen3-235B as an auxiliary model. For OpenAI Deep Research, we collect reports generated from the service environment as of August 2025.

## H. Human-Correlation Experiment Setup

We measure alignment between LLM judges and expert human judgments on 45 reports. We consider five domains and sample three tasks from each, yielding 15 tasks in total. For each task, we randomly sample three reports from those produced by five Deep Research systems—OpenAI Deep Research, Gemini 2.5 Pro Deep Research, Claude Opus 4.1 Deep Research, WebThinker, and Qwen3-235B Deep Research—resulting in $15 \times 3 = 45$ reports overall. WebThinker (Li et al., 2025b) uses Qwen3-235B as an auxiliary model.
Each report was independently evaluated by two domain experts matched to the report's topic, supporting both LLM–human correlation and human–human reliability analyses (90 ratings total; 45 reports × 2 experts). Experts had at least a master's degree in a relevant field or comparable professional experience and, taking various factors into account, assigned an overall report quality score on a 1–5 scale (fractional values allowed). We used the mean of the two expert ratings as the report-level human score for LLM–human correlation, and we also reported agreement between the two experts' ratings. Evaluating a report took 1.5 hours on average. Compensation followed the vendor's standard payment framework, and we verified adherence to relevant procedures and policies.
We measure correlation with human judgments for five evaluator models (GPT-5.2, GPT-5-mini, Claude Opus 4.1, Claude Sonnet 4.5, and Gemini 2.5 Pro). Each model evaluates the same set of reports under each evaluation setting (Vanilla,

+Dimensions, +Granular Rubrics, +Expert Guidance). Under the +Dimensions setting, evaluation is performed at the level of five report-quality dimensions. Under +Granular Rubrics and +Expert Guidance, evaluation is performed down to the level of rubric items that further break down those dimensions.

To measure alignment with expert human judgments, we report Pearson correlation ($r$), Spearman rank correlation ($\rho$), and pairwise agreement (PA). Each task corresponds to a single query and contains three reports. For each task, we compute Pearson $r$ and Spearman $\rho$ between human and model scores across the three reports, and report the mean over the 15 tasks. Following DeepBench (Du et al., 2026), PA is the fraction of report pairs within a task for which the model's relative preference matches the human relative preference; with three reports, each task has $\binom{3}{2} = 3$ pairs. We compute PA per task and report the mean across tasks. To match the human 1–5 scale, we rescale model scores to 1–5 by dividing 1–10 scores by 2. For PA, a pair is counted as an agreement if the model and human judgments induce the same relation ($>$, $<$, or $=$) for that pair. Tasks with undefined correlations (e.g., zero variance) are excluded from the corresponding averages.

For each setting, we compute the three metrics ($r$, $\rho$, and PA) independently for each of the five evaluator models, and then report the average across the models.

Expert contributors participated as paid contractors via a professional vendor; no sensitive personal data were collected, and participation followed the vendor's consent and compensation policies.

## I. Judge Backbone Selection

To select the judge models, we conduct backbone ablations comparing the performance and cost of several candidate backbones.

For Report Quality Assessment, we measure performance by correlation with human evaluators. Using the same setup as Table 3 and Appendix H, we apply five candidate judge backbones to the same set of 45 reports and compute Pearson correlation, Spearman correlation, and pairwise agreement. As shown in Table 17, GPT-5.2 achieves the highest performance across all three metrics. We therefore use GPT-5.2 for Report Quality Assessment.

For Information Verification, we compare the performance and cost of candidate backbones for claim extraction/classification and claim verification. Since this module checks each claim against external documents, it requires a large number of API calls, making cost especially important. As shown in Tables 18 and 19, GPT-5-mini achieves the best or competitive performance in both stages while maintaining low cost. We therefore use GPT-5-mini for the Information Verification module.

## J. Additional Ablation for Report Quality Assessment

Following the human-correlation setup in Section 6.3, we conduct a leave-one-component-out ablation for Report Quality Assessment. While Table 3 evaluates components by incrementally adding them, this analysis measures how human alignment changes when each component is removed from the final Report Quality Assessment setting. In the DEER – Granular Rubrics setting, we remove the shared granular rubrics while keeping task-specific expert guidance. In the DEER – Expert Guidance setting, we remove task-specific expert guidance while keeping the shared granular rubrics.

As shown in Table 20, removing either the shared granular rubrics or expert guidance decreases Pearson correlation, Spearman correlation, and pairwise agreement. This suggests that the two components play complementary roles in Report Quality Assessment.

## K. Ranking Stability Across Evaluator Models

Table 4 shows score-level consistency across different evaluator models, but does not directly measure whether they produce similar rankings over the same set of reports. We therefore conduct an additional ranking-stability analysis.

For Report Quality Assessment, we use the same 45 reports and five evaluator models as in Table 4. For each evaluator, we convert its scores over the 45 reports into a ranking, and measure agreement among the five evaluator rankings using Kendall's W. We also compute Spearman's $\rho$ and Kendall's $\tau$ for each pair of evaluator models over the same 45-report score vectors, and report the average across evaluator pairs.

As shown in Table 21, the final proposed method, +Expert Guidance, achieves the highest ranking agreement across Kendall's W, average pairwise Spearman's $\rho$, and average pairwise Kendall's $\tau$. This indicates that the proposed method shows the highest ranking stability under evaluator model changes among the compared settings.

For Information Verification, we compare three backbone choices for the Information Verification pipeline—GPT-5-mini, Gemini 2.5 Flash, and Claude Haiku 4.5—on 90 reports in CS/AI. Using the final Integrity and Sufficiency scores produced by each backbone, we compute report ranking agreement in the same manner.

As shown in Table 22, ranking agreement remains high for both Integrity and Sufficiency. Kendall's W is 0.934 and 0.936, and average pairwise Spearman's $\rho$ is 0.901 and 0.904, respectively. This indicates that the Information Verification pipeline

also shows stable ranking agreement across backbone choices.

## L. Scoring Design Analysis

We conduct an ablation to examine the effect of scoring design on human alignment under the same human-correlation setup. We compare three variants: DEER's 5-band anchor + 10-point scale, a 5-band anchor + 5-point scale, and a no-anchor + 10-point scale.

Table 23 reports the human-correlation results for each scoring design. DEER's 5-band anchor + 10-point scale achieves the highest alignment across Pearson correlation, Spearman correlation, and pairwise agreement.

## M. Length Bias Analysis

Prior work has shown that LLM judges can exhibit length bias when evaluating long-form generations. To examine whether the DEER report-quality scores are affected by report length, we conduct an additional length-bias analysis over all 800 baseline reports used in our main evaluation. For each report, we measure its length by word count and compute its average score across the five report-quality dimensions. We then measure the association between report length and evaluation score using both Pearson correlation, which captures linear association, and Spearman correlation, which captures monotonic rank association.

Table 24 reports the overall correlation computed across all 800 reports. The Pearson correlation is nearly zero ($r = -0.02$, $p = 0.67$), and the Spearman correlation is also very small ($\rho = 0.10$, $p < 0.01$). Although the Spearman correlation is statistically significant due to the large sample size, its magnitude is negligible. This suggests that the DEER report-quality scores are not primarily driven by longer outputs.

We further examine whether the near-zero aggregate correlation masks task-specific length bias. In particular, the overall correlation could be small if some tasks favor longer reports while others favor shorter reports. To test this, we compute a separate correlation for each of the 50 tasks, using the 16 baseline reports generated for that task. Table 25 summarizes the resulting distribution of 50 task-level correlations. Most task-level correlations fall in the near-zero interval $[-0.2, 0.2]$: 36 out of 50 tasks for Pearson correlation and 32 out of 50 tasks for Spearman correlation. The remaining larger correlations are relatively rare and appear in both positive and negative directions. These results show that the near-zero overall correlation is not merely due to positive and negative task-level correlations canceling each other out. Instead, report length has little association with evaluation scores for most tasks, suggesting that DEER's report-quality evaluation is not substantially driven by output length.

## N. Prompts

This appendix provides the prompt templates used in our evaluation pipeline. Figure 8 shows a condensed evaluator prompt template for the Request Fulfillment dimension, one of the report-quality dimensions. Figure 9 shows the full prompt used for the claim extraction and classification task. Figure 10 shows the complete prompt for the claim verification and source reliability task.

**Example Expert Evaluation Guidance (MDP / Value Iteration)**

1. **Formal MDP and Value Iteration Setup**
   The report precisely defines the components of a finite-state MDP (state space, action space, transition probabilities, reward function, discount factor $\gamma$), states the Bellman optimality equation, and presents the value iteration update rule with consistent notation (e.g., $V_{k+1} = \mathcal{T}V_k$), ensuring all subsequent analysis refers unambiguously to this formal framework.

2. **Contraction Mapping and Norm Specification**
   The report identifies a complete metric space (e.g., bounded real-valued functions on states) and proves that the Bellman optimality operator is a contraction mapping under the sup-norm or an equivalent norm, explicitly deriving the contraction modulus as $\gamma$ and showing that $||\mathcal{T}V - \mathcal{T}V'||_\infty \leq \gamma||V - V'||_\infty$ for all value functions $V, V'$.

3. **Banach Fixed-Point Theorem Application**
   The report invokes the Banach fixed-point theorem to establish existence, uniqueness, and geometric convergence of the optimal value function, clearly stating the theorem's conditions (completeness of space, contraction property) and demonstrating how they are satisfied by the MDP and value iteration operator.

4. **Reward Boundedness and Sign Independence**
   The report analyzes the role of reward boundedness in ensuring geometric convergence, demonstrating that bounded rewards (regardless of sign or symmetry) preserve contraction under the Bellman operator, and proving that unbounded rewards can violate the contraction condition; it explicitly states that the sign or symmetry of the reward interval (e.g., symmetric around zero or non-negative) does not affect convergence as long as boundedness and discounting are maintained.

5. **Geometric Convergence Rate Derivation**
   The report derives the geometric convergence rate in terms of $\gamma$ and reward bounds, providing a tight bound on $||V_k - V^*||_\infty \leq \frac{\gamma^k}{1-\gamma}||V_1 - V_0||_\infty$, and explains how the effective rate depends on $\gamma$, not directly on reward values unless they influence the norm or stopping condition.

6. **Edge Case Analysis for Reward Boundedness**
   The report analyzes representative edge cases for reward boundedness, including symmetric vs. asymmetric intervals, zero rewards, and unbounded or improperly scaled rewards, using theoretical reasoning or controlled simulations to test whether geometric convergence holds or breaks down under each condition.

7. **Impact of Reward Sign and Symmetry**
   The report evaluates whether the sign or symmetry of the reward interval (e.g., symmetric around zero vs. non-negative) affects convergence, proving that geometric convergence depends only on the discount factor and boundedness, not on sign, unless reward shaping alters the effective $\gamma$ or violates boundedness.

8. **Numerical Stability and Termination Criteria**
   The report analyzes practical convergence behavior, including how finite precision arithmetic, reward scaling, and choice of stopping criterion (e.g., $||V_{k+1} - V_k||_\infty < \epsilon$) interact with theoretical guarantees, and demonstrates at least one case where numerical error or poor scaling leads to premature termination or slow apparent convergence.

9. **Counterexamples for Non-Convergence**
   The report constructs at least one verifiable scenario where value iteration fails to converge geometrically—such as when $\gamma = 1$, rewards are unbounded, or the contraction condition is violated—and explains how this informs the boundary of the reward range for guaranteed convergence.

10. **Discount Factor and Reward Interaction**
    The report examines how the interplay between $\gamma$ and reward magnitude influences the effective contraction rate, showing that while $\gamma$ controls the rate, reward scaling affects the constant factor in the convergence bound, and that rescaling rewards (e.g., dividing by $R_{\max}$) can normalize behavior across different domains.

11. **Illustrative Examples with Reproducible Design**
    The report includes illustrative examples that feature explicit rules for MDP construction (number of states/actions, transition sparsity, reward assignment), initialization of $V_0$, value of $\gamma$, stopping threshold, and random seed control. These examples should be described with enough detail to make the reasoning transparent and verifiable.

12. **Limitations and Generalizations**
    The report discusses limitations of the geometric convergence guarantee, including infinite state spaces, continuous actions, non-stationary environments, or non-linear function approximation, and clarifies whether the reward boundedness condition extends to policy iteration, Q-learning, or other dynamic programming variants, grounding each claim in earlier analysis.

*Figure 6.* Example of Expert Evaluation Guidance

| Domain (80) | Reference |
|---|---|
| AI (8) | National Institute of Standards and Technology, 2023; Stanford HAI, 2025; NeurIPS Foundation, 2025; International Conference on Machine Learning, 2025; Springer Nature, 2025; ACL Rolling Review author guidelines[a]; ICLR 2024 Author Guide[b]; JMLR author instructions[c] |
| Biology (4) | Page et al., 2021; Schulz et al., 2010; International Society for Stem Cell Research, 2025; Percie du Sert et al., 2020 |
| Business (4) | Global Reporting Initiative, 2023; Garner, 2013; IFRS Foundation, 2021; ISO 30414:2018[d] |
| Chemistry (4) | American Chemical Society, 2021; International Union of Pure and Applied Chemistry, 2007; The Journal of Organic Chemistry, 2025; National Research Council, 2011 |
| Computer Science (4) | International Organization for Standardization, 2011; Association for Computing Machinery, 2025; IEEE Computer Society, 2018; Wilkinson et al., 2016 |
| Earth & Env. Science (4) | IPCC, 2019; U.S. Geological Survey, 2024; Eaton et al., 2024; Ecological Society of America, 2021 |
| Economics (4) | Chang et al., 2024; American Economic Association, 2024; The Econometric Society, 2024; OECD, 2011 |
| Education (4) | What Works Clearinghouse, 2022; American Educational Research Association, 2006; Institute of Education Sciences, 2022; American Educational Research Association et al., 2014 |
| Engineering (4) | IEEE Computer Society, 2025; IEEE, 2025; NASA, 2016; IEEE Computer Society, 2018 |
| Finance (4) | International Accounting Standards Board, 2024; CFA Institute, 2024; 2020; U.S. Securities and Exchange Commission, 2024 |
| History (4) | American Historical Association, 2024; University of Chicago Press, 2024; Oral History Association, 2024; Organization of American Historians, 2018 |
| Law (4) | Columbia Law Review et al., 2020; Garner, 2019; U.S. Congress, 2024; American Bar Association, 2023 |
| Linguistics (4) | Linguistic Society of America, 2024; Comrie, B. and Haspelmath, M. and Bickel, B., 2015; Berez-Kroeker et al., 2018; International Phonetic Association, 1999 |
| Mathematics (4) | European Mathematical Society, 2025; American Mathematical Society, 2022; Society for Industrial and Applied Mathematics, 2024; National Institute of Standards and Technology, 2024 |
| Medicine (4) | World Medical Association, 2013; von Elm et al., 2007; Gagnier et al., 2013; Chan et al., 2013 |
| Philosophy (4) | American Philosophical Association, 2024; Stanford Encyclopedia of Philosophy, 2025; British Philosophical Association, 2024; Australasian Association of Philosophy, 2023 |
| Physics (4) | American Physical Society, 2023; International Union of Pure and Applied Physics, 2010; Navas et al., 2024; Mohr et al., 2024 |
| Political Science (4) | American Political Science Association, 2018; OECD, 2020; American Political Science Association, 2022; DA-RT principles[e] |
| Psychology (4) | American Psychological Association, 2025; Center for Open Science, 2024; British Psychological Society, 2021; American Educational Research Association et al., 2014 |
| Sociology (4) | American Association for Public Opinion Research, 2021; Tong et al., 2007; National Commission for the Protection of Human Subjects of Biomedical and Behavioral Research, 1979; American Sociological Association, 2022 |

*Table 11.* References by domain used in the Deep Research Report Evaluation Taxonomy. The table contains 80 listings from 78 unique sources, with two interdisciplinary standards cross-listed.

[a] ACL checklist: `https://aclrollingreview.org/responsibleNLPresearch/`
[b] ICLR Author Guide: `https://iclr.cc/Conferences/2024/AuthorGuide`
[c] JMLR instructions: `https://jmlr.org/author-info.html`
[d] ISO 30414:2018: `https://www.iso.org/standard/69338.html`
[e] DA-RT principles: `https://www.dartstatement.org/`

| Dim / Sub-dim | Criterion / Rubric items Description |
| --- | --- |
| 1. Request Fulfillment
└ 1.1 Completeness | **1.1.1 Criterion** Does the report include all required elements without omission and present each clearly?
└ **[1.1.1.1 (Coverage)]** The report must include all elements required by the User Query and EG, and each element must be clearly explained; any element falling short of the EG standard is treated as omitted.
└ **[1.1.1.2 (Coverage)]** Each major requirement from the User Query must be developed with sufficient length and substantive, EG-relevant explanation; filler content does not satisfy the requirement.
└ **[1.1.1.3 (Quality)]** Each required element must be supported by appropriate EG-guided evidence, reasoning, and validation; materially EG-inconsistent reasoning or validation is treated as analytically unsound.
└ **[1.1.1.4 (Quality)]** Only elements satisfying the soundness requirement are evaluated for depth; they must be sufficiently supported and developed in accordance with the EG, while unsound elements are treated as insufficient. |
| 2. Analytical Soundness
└ 2.2 Reasoning | **2.2.5 Criterion** Are all claims logically derived from previously presented facts, data, interpretations, and reasoning, without logical leaps or missing steps?
└ **[2.2.5.1 (Coverage)]** All claims requiring support must be validly inferred from factually correct and EG-consistent premises; claims based on inaccurate, inconsistent, or missing bases are treated as logical leaps.
└ **[2.2.5.2 (Coverage)]** Major evidence necessary for logical development, especially EG-specified core evidence, must be included without omission.
└ **[2.2.5.3 (Quality)]** Links between claims, evidence, and reasoning must be clear, robust, and well developed; major claims must be supported by both specific evidence and the overall body of evidence. |
| 3. Structural Coherence
└ 3.1 Introduction | **3.1.1 Criterion** Does the introduction clearly present the report's topic, problem, and significance without excessive generalization, while providing sufficient context and motivation for the reader?
└ **[3.1.1.1 (Coverage)]** The introduction must include the report's topic, problem, and significance, along with sufficient background and motivation for readers to understand the report's context and rationale.
└ **[3.1.1.2 (Quality)]** The introduction must be sufficiently developed for a professional report, with adequate length and coverage of all required components.
└ **[3.1.1.3 (Quality)]** Each component must be described clearly and specifically, without excessive generalization or ambiguity.
└ **[3.1.1.4 (Quality)]** The introduction must present its components in a logical and coherent flow, allowing the reader to easily grasp the report's overall direction. |
| 4. Format & Style
└ 4.2 Writing Quality | **4.2.3 Criterion** Are technical terms defined when they first appear and used consistently thereafter?
└ **[4.2.3.1 (Coverage)]** Technical terms and field-specific concepts must be clearly defined when they are central, ambiguous, specialized, non-standard, or not guaranteed to be known by the intended audience; standard terms do not require formal definitions if clear from context.
└ **[4.2.3.2 (Coverage)]** After being defined, technical terms, abbreviations, and symbols must be used consistently with their definitions throughout the document. |
| 5. Ethics & Compliance
└ 5.2 Safety & Impact | **5.2.1 Criterion** Are the potential impacts of proposed policies, technologies, strategies, or research outcomes sufficiently considered, including key implications, side-effects, and interpretations from multiple perspectives where essential?
└ **[5.2.1.1 (Coverage)]** Potential side-effects or limitations must be discussed where essential.
└ **[5.2.1.2 (Coverage)]** Multiple stakeholder perspectives or contextual viewpoints must be included where essential.
└ **[5.2.1.3 (Quality)]** Key implications must be presented in a balanced way, with relevant contexts sufficiently considered where essential.
└ **[5.2.1.4 (Quality)]** Each identified impact must be analyzed with adequate detail and supported by data, evidence, or clear reasoning where essential. |

*Table 12.* Example criteria and rubric items from a four-level taxonomy (dimension, sub-dimension, criterion, and item: Coverage or Quality). Items are abbreviated for readability while preserving the main operational criteria.

| Score Range | Coverage | Quality |
|---|---|---|
| 9–10 (Perfect) | Fully meets all requirements; no omissions; no revision needed | Excellent quality in all relevant aspects; no revision needed—top-tier international journal level, or high-end professional report meeting or exceeding standards in specific technical/industrial contexts |
| 7–8 (Excellent) | Meets almost all requirements; only 1–2 minor omissions, minimal impact | High quality; meets most academic/professional standards, only minor improvements possible—solid peer-reviewed journal, excellent doctoral research, high-quality industry report level |
| 5–6 (Good) | Meets more than half; meets most key requirements, minor elements missing | Meets essential professional standards; clear structure and competent analysis but room for improvement—well-written master's thesis or standard professional report level |
| 3–4 (Inadequate) | Partially meets; several key omissions | Noticeable flaws in several aspects; significant revision needed—undergraduate thesis or entry-level professional report level |
| 1–2 (Poor) | Most requirements are missing or treated only superficially | Fails to meet basic professional standards; lacks depth, rigor, accuracy—below undergraduate level; unsuitable for publication or professional use |

*Table 13.* Interpretation of 1–10 score ranges for Coverage (C) and Quality (Q) factors.

| Type | Definition and Example |
|---|---|
| A | **Cited Claim**: A claim explicitly including a citation marker within the sentence. *Example:* "Multi-junction solar cells achieve efficiencies above 45% [1]." |
| B | **Uncited – Same Section / Paragraph**: When the evidence citation exists in a previous sentence within the same section (or paragraph). *Example:* "This efficiency improvement is due to the layered structure." (Evidence $\rightarrow$ L1.S1) |
| C | **Uncited – Previous Section / Paragraph**: When the citation evidence exists in a previous section or paragraph. *Example:* "These findings confirm the results of earlier solar-cell studies." (Evidence $\rightarrow$ L1.S2) |
| D | **Uncited – Structural Recap**: A claim corresponding to a restatement of content in the document structure, such as introduction, conclusion, or summary. *Example:* "In conclusion, this paper reviewed recent advances in solar technology." |
| E | **Uncited – No Citation Required**: The author's direct results, general knowledge, or a claim not requiring citation. *Example:* "Photosynthesis converts light energy into chemical energy." |
| F | **No Citation – Unknown Source**: A claim requiring external evidence but for which no source is presented. *Example:* "These panels can last for 50 years without degradation." |

*Table 14.* Information Verification Module A–F Claim Type Definitions and Examples. Types A–C are targets for external evidence verification, while Types D–E–F are classified as internal information or unverifiable areas.

| Method | Jaccard | Precision | Recall |
|---|---|---|---|
| **Backtracking (Ours)** | **0.7070** | **0.7109** | 0.7383 |
| Sliding ($k = 5$) | 0.6822 | 0.6822 | 0.8022 |
| Sliding ($k = 10$) | 0.6247 | 0.6247 | 0.8791 |
| Sliding ($k = 15$) | 0.5627 | 0.5627 | **0.9341** |

*Table 15.* Baseline comparison on correctly predicted B/C claims ($N = 131$). Backtracking achieves the best balance of precision, recall, and Jaccard.

---

**Simplified Batch Extraction Prompt Structure**

**System:** You are an expert fact-checker and claim extractor.
**User:**
# Full Report Context
{Full_Report_Text}

# Target Sentences to Extract Claims From
L10.S1: ...
...
L10.S20: ...

Extract claims only from the target sentences above. Use the full report context for coreference resolution.

---

**Example JSON Output**

```
{
 "claims": [{
  "position": "L10.S1",
  "index": 1,
  "claim_text": "Solar efficiency reached
  45% [1].",
  "claim_class": "A",
  "direct_citation": "[1]",
  "evidence_position": null
 }, ...]
}
```

*Figure 7.* Simplified batch extraction prompt structure and an example JSON output.

| | | | | | Original | | Adversarial | |
| Batch | Effort | Embed. | Top-K | Cost/1k ($) | Acc (%) | F1 | Acc (%) | F1 |
|---|---|---|---|---|---|---|---|---|
| 10 | high | BM25 | 4 | 3.61 | 77.01 | 87.25 | 87.36 | 85.16 |
| 10 | high | OpenAI | 4 | 3.67 | 78.16 | 88.00 | 88.46 | 86.62 |
| 10 | low | BM25 | 4 | 0.91 | 79.31 | 88.00 | 87.36 | 85.16 |
| 10 | low | OpenAI | 4 | 1.03 | 78.16 | 88.00 | 89.01 | 87.34 |
| 10 | medium | BM25 | 4 | 1.75 | 73.56 | 84.93 | 87.36 | 85.16 |
| 10 | medium | OpenAI | 4 | 1.62 | 73.56 | 84.93 | 87.91 | 85.90 |
| 20 | high | BM25 | 2 | 3.08 | 79.31 | 88.16 | 88.46 | 86.79 |
| 20 | high | BM25 | 4 | 3.38 | 79.31 | 88.74 | 89.56 | 88.05 |
| 20 | high | OpenAI | 2 | 3.34 | 77.01 | 87.25 | 85.71 | 83.33 |
| 20 | high | OpenAI | 4 | 3.31 | 75.86 | 86.49 | 88.46 | 86.79 |
| 20 | low | BM25 | 2 | 1.02 | 73.56 | 84.93 | 90.11 | 88.61 |
| 20 | low | BM25 | 4 | 1.30 | 81.61 | 90.20 | 89.01 | 87.34 |
| 20 | low | OpenAI | 2 | 0.95 | 77.01 | 87.25 | 88.46 | 86.79 |
| 20 | low | OpenAI | 4 | 1.28 | 79.31 | 88.74 | 89.56 | 87.90 |
| 20 | medium | BM25 | 2 | 1.57 | 74.71 | 85.71 | 86.81 | 84.42 |
| 20 | medium | BM25 | 4 | 1.79 | 79.31 | 88.74 | 88.46 | 86.62 |
| 20 | medium | OpenAI | 2 | 1.43 | 73.56 | 84.93 | 87.91 | 86.08 |
| 20 | medium | OpenAI | 4 | 1.71 | 74.71 | 85.71 | 88.46 | 86.79 |

*Table 16.* Ablation study of GPT-5-mini on Context Retrieval setting. Showing the impact of Batch Size, Reasoning Effort, Embedding Method, and Top-K chunks on performance. **Original** refers to the standard dataset, and **Adversarial** refers to the augmented dataset.

| Model | Pearson | Spearman | Pairwise | Cost In | Cost Out |
|---|---|---|---|---|---|
| GPT-5.2 | 0.81 | 0.80 | 0.89 | 1.75 | 14.00 |
| Gemini 2.5 Pro | 0.76 | 0.70 | 0.82 | 1.25–2.50 | 10.00–15.00 |
| GPT-5-mini | 0.75 | 0.73 | 0.87 | 0.25 | 2.00 |
| Claude Opus 4.1 | 0.63 | 0.67 | 0.82 | 15.00 | 75.00 |
| Claude Sonnet 4.5 | 0.73 | 0.63 | 0.80 | 3.00 | 15.00 |

*Table 17.* Judge-backbone ablation for Report Quality Assessment. Costs are reported per million input and output tokens.

| Model | Recall | Cls. F1 | Cost |
|---|---|---|---|
| GPT-5-mini | 92.1 | 68.8 | 0.16 |
| GPT-5.2 | 96.3 | 59.3 | 1.66 |
| Claude Haiku 4.5 | 91.9 | 65.8 | 1.45 |
| Gemini 2.5 Flash | 91.2 | 60.1 | 0.44 |

*Table 18.* Judge-backbone ablation for claim extraction and classification. Cost is reported for the evaluated set.

| Model | F1 | Cost |
|---|---|---|
| GPT-5-mini | 87.3 | 0.95 |
| Gemini 2.5 Flash | 86.3 | 1.28 |
| Claude Haiku 4.5 | 86.7 | 2.87 |

*Table 19.* Judge-backbone ablation for claim verification. Cost is reported for the evaluated set.

| Method | Pearson | Spearman | Pairwise |
|---|---|---|---|
| DEER – Granular Rubrics | 0.631 | 0.598 | 0.764 |
| DEER – Expert Guidance | 0.643 | 0.587 | 0.760 |
| DEER | 0.734 | 0.707 | 0.840 |

*Table 20.* Leave-one-component-out ablation for Report Quality Assessment.

| Method | Kendall's W | Avg. Pairwise Spearman's $\rho$ | Avg. Pairwise Kendall's $\tau$ |
|---|---|---|---|
| Vanilla | 0.659 | 0.577 | 0.527 |
| + Dimensions | 0.664 | 0.578 | 0.495 |
| + Granular Rubrics | 0.583 | 0.480 | 0.432 |
| + Expert Guidance | 0.742 | 0.678 | 0.558 |

*Table 21.* Ranking stability across evaluator models for Report Quality Assessment. Each row corresponds to the same component-added evaluation setting as in Table 4, and rankings are computed using the average score over the five report-quality dimensions.

| Dimension | Kendall's W | Avg. Pairwise Spearman's $\rho$ | Avg. Pairwise Kendall's $\tau$ |
|---|---|---|---|
| Integrity | 0.934 | 0.901 | 0.747 |
| Sufficiency | 0.936 | 0.904 | 0.741 |

*Table 22.* Ranking stability across backbone choices for the Information Verification pipeline. Rankings are computed from the final Integrity and Sufficiency scores produced by each backbone.

| Setting | Pearson | Spearman | Pairwise |
|---|---|---|---|
| 5-band anchor + 10-point scale (Ours) | 0.734 | 0.707 | 0.840 |
| 5-band anchor + 5-point scale | 0.694 | 0.660 | 0.809 |
| No-anchor + 10-point scale | 0.722 | 0.633 | 0.796 |

*Table 23.* Ablation of anchor and score-range variants for Report Quality Assessment.

| Metric | Pearson $r$ | $p$ | Spearman $\rho$ | $p$ |
|---|---|---|---|---|
| Avg. score | $-0.02$ | 0.67 | 0.10 | $< 0.01$ |

*Table 24.* Correlation between report length and average report-quality score across all 800 baseline reports.

| Correlation range | Pearson (# Tasks) | Spearman (# Tasks) |
|---|---|---|
| $[-0.6, -0.4)$ | 1 | 1 |
| $[-0.4, -0.2)$ | 3 | 7 |
| $[-0.2, 0.2)$ | 36 | 32 |
| $[0.2, 0.4)$ | 6 | 7 |
| $[0.4, 0.6)$ | 4 | 3 |

*Table 25.* Distribution of task-level correlations between report length and average report-quality score across 50 tasks. Each task-level correlation is computed over the 16 baseline reports generated for that task.

**Evaluation Prompt Template for Request Fulfillment**

```
SYSTEM_PROMPT = """
You are an expert evaluator for expert-level long-form professional reports. Evaluate and score the provided report item by item using
the provided rubric and the Expert Evaluation Guidance (EG).

# 1. Overview
Evaluate the report using the provided rubric, which operationalizes the requirements of a professional, expert-level long-form report.
"User Request" is the original writing request used to generate the report.
"Report to Evaluate" is the long-form report evaluated against the User Request.
For each rubric item (e.g., C1-1, C1-2, Q1-1, Q1-2), provide systematic
evaluation reasoning with relevant report evidence and assign an item-level score.
Scores must be integers from 1 to 10. If no assessable material exists for an item, enter "N/A".

# 2. Evaluation Method
Evaluate each rubric item strictly using the provided rubric and EG.
Do not make arbitrary judgments outside the rubric and EG.

# 3. Expert Evaluation Guideline (EG)
The EG provides task-specific expert criteria as concrete, verifiable statements.
The EG has absolute priority: each rubric item must be evaluated with all applicable EG requirements applied in full.
If EG requirements are not met, no high score (Perfect or Excellent) may be awarded regardless of supplementary strengths.

# 4. Rubric Items
[Omitted for brevity.]

# 5. How to Score: Coverage (C) vs. Quality (Q)
Each rubric item has either a Coverage (C) or Quality (Q) attribute, and each is evaluated independently.

Coverage evaluates whether every required component is present and fully addressed.
The evaluator identifies required elements, checks each as Pass/Fail, classifies failures as core gaps or minor omissions, and assigns
a 1-10 score.
[Detailed Coverage scoring bands omitted for brevity; see Table 12.]

Quality evaluates how well the report executes the relevant written content.
The evaluator considers only what is written, makes an academic/professional quality judgment, and lowers the score if any core element
falls short.
[Detailed Quality scoring bands omitted for brevity; see Table 12.]

Core principles:
* Even one core gap makes Excellent impossible for Coverage.
* Multiple core gaps make Good impossible for Coverage.
* If an EG core element falls short, the Quality score should be lowered.

# 6. Output Format
For each rubric item, write the "description" as score justification grounded in the scoring guidelines, not as a general pros/cons summary.
The description must match the assigned score band, identify concrete deficiencies when relevant, use report evidence, and keep deficiency
severity consistent with the score.
[JSON schema omitted for brevity.]
"""

USER_PROMPT = """
[User Request]
{query}

[Report to Evaluate]
{doc}

[Expert Evaluation Guidance (EG)]
{core_criteria}
"""
```

*Figure 8.* Condensed evaluator prompt template for Request Fulfillment. The template preserves the system–user prompt structure, EG-priority rule, Coverage/Quality scoring logic, and output-format requirements, while omitting the full rubric item list, detailed scoring bands, and JSON output schema for space.

**Full Prompt for Claim Extraction and Classification**

You are an expert fact-checker. Your task is to extract distinct claims from the provided Target Sentences
and classify each one into one of the following categories.

Class Definitions:

| Class | Definition |
| --- | --- |
| A: Cited Claim | Claims that include explicit citations (e.g., [1] or (Kim, 2024)) directly within the sentence. |
| B: Uncited - Same Section/Paragraph | Claims whose supporting citations appear within the same section or paragraph. Crucially, the Evidence Position MUST be in the same section or paragraph. If the evidence is in a previous section, it is Class C. |
| C: Uncited - Previous Section/Paragraph | Claims whose supporting citations appear in previous sections or paragraphs. T1 he Evidence Position MUST be in a previous section. |
| D: Uncited - Recap/Structural | Simple mentions of introduction, conclusion, abstract, or structural summaries. |
| E: Uncited - Citation not Required | Widely accepted facts, common knowledge, author's own findings, subjective claims, experimental results, methodological contributions, etc. WARNING: Conclusions, results, or findings of the paper are NOT Class E. They are specific claims that require evidence (Class A/B/C) or are uncited (Class F). |
| F: No-Citation - Unknown Source | Specific factual assertions that require verification but lack any identifiable supporting source in the text. If a claim is specific (e.g., "X has a radius of Y", "X is planar") and has no citation attached to it or the sentence immediately preceding/following it that covers this fact, it is Class F. Do not assume a nearby citation covers it unless it clearly does. |

Instructions:
1. Read the Report Context to understand the global context.
2. Process the "Target Sentences":
   - Break down the text into atomic claims. A single sentence may contain multiple claims
   (e.g., "X is Y, and Z requires W" -> Claim 1: "X is Y", Claim 2: "Z requires W").
   - Extract ALL statements, including facts, opinions, structural descriptions, and summaries.
3. For each extracted claim, analyze its relationship with the context and citations:
   - Step 1: Specificity Check. Contains numbers, chemical properties, specific results? -> Likely A, B, C, or F.
   - Step 2: Citation Check.
     - Citation in same sentence? -> Class A.
     - Citation in same paragraph? -> Class B.
     - Citation in previous section? -> Class C.
   - Step 3: Uncited Check.
     - Specific fact but no citation? -> Class F.
     - General knowledge/methodology/author opinion? -> Class E.
     - Structural summary? -> Class D.
4. Determine Evidence Position:
   - For Class B or C, identify the exact sentence index (e.g., "L1.S1") that contains the citation supporting this claim.
   - Must contain an explicit citation.
5. Output Format:
   - Return a JSON object with a list of claims.
   - Each claim must include: `position` (line/sent index from input), `claim` (text), `claim_type` (A-F), `rationale`,
   `numeric` (bool), `citations` (list of strings), `implicit_citations` (list), `cross_references` (list).

Input Format:
# Report Excerpt
...
# Target Sentences
L1.S1: ...
L1.S2: ...

Extraction and Classification Logic:
- If a sentence is "Most perovskites are unstable[1], but our new material is stable.", extract TWO claims:
  1. "Most perovskites are unstable." (Class A, citations=['1'])
  2. "Our new material is stable." (Class E - Author's finding, or Class F if it lacks proof provided elsewhere)

Examples:

*Example Input:*
L1.S1: Several studies[1] have shown that urban green spaces can reduce ambient air temperatures by up to 2 °C. This is crucial.

*Example Output (Conceptual):*
1. Claim: "Several studies have shown that urban green spaces can reduce ambient air temperatures by up to 2 °C."
   - Class: A
   - Citations: ["1"]
   - Position: "L1.S1"
2. Claim: "This is crucial."
   - Class: B (supported by L1.S1)
   - Evidence Position: "L1.S1"
   - Position: "L1.S1"

*Figure 9.* Full prompt for Claim Extraction and Classification.

**Full Prompt for Claim Verification**

```
You are an expert fact-checker. Verify the following claims against the provided context.
Be extremely strict. High precision is required.

For each claim, determine if it is supported by the context.
Result should be based on semantic meaning, not just keyword matching. If the text implies the claim is true,
mark it as supported.

Verification Result (`result`):
- supported: The text explicitly states or clearly implies the claim is true.
- conflict: The text explicitly contradicts the claim.
- not_supported: The text does not contain enough information.
- error: Access/processing error (4xx/5xx, captcha, paywall, etc.)

[Error Cases — Details]
- Set result to "error" in the following cases:
  - HTTP 4xx/5xx errors
  - Access restrictions such as Captcha, Paywall, Authentication Required
  - Other processing error messages
- Briefly describe the specific error cause in explanation (e.g., "404 Not Found", "paywall restriction").

[Document Reliability — Top-Level Fields]
- reliable (true/false):
  - Evaluate whether the entire URL is from a trustworthy source
  - Example 1: Official statistics, academic journals, authoritative institutions -> true
  - Example 2: Personal blogs, social media posts, etc. -> false
- reliable_explanation (string or null):
  - Rationale for reliability judgment (optional)

# Examples:
Claim: "GA+ prevents degradation."
Context: "Guanidinium (GA+) ... helps passivate defects, stabilizing the structure against degradation."
Result: supported

# url: {url}

# claims:
{claims}

# Context:
{context}
```

*Figure 10.* Full prompt for Claim Verification.

