# OpenReview forum: "DEER: A Benchmark for Evaluating Deep Research Agents on Expert Report Generation"
_ICML.cc/2026/Conference — ICML 2026 regular_

### Official Review · Reviewer_fGXM · 2026-03-05

**Soundness:** 3
**Presentation:** 3
**Significance:** 3
**Originality:** 2
**Overall Recommendation:** 4
**Confidence:** 4

**Summary:**

This paper proposes a novel evaluation benchmark called DEER (built on top of HLE tasks) used to evaluate deep research tasks on a wide variety of subjective metrics. Their primary contributions include creation of an expert designed taxonomy for open ended evaluation of output reports, utilizing expert guidance for evaluation and effective claim extraction and classification. Through their experiments, the authors compare LLM judge performance on several domains of tasks from HLE and further perform human evaluation of their approach.

**Compliance With Llm Reviewing Policy:**

Affirmed.

**Final Justification:**

The final rebuttal has partially resolved my doubts. I, however, still feel like the rubric scale is not well defined and the authors do not understand the scale that they have themselves defined (for example, what is the difference between a score of 5 and 6 on a scale of 10?). This severely contaminates not just the paper results but also the human evaluation studies since human annotators do not understand the differences exactly. The original judgment was made on the basis of the initial submitted paper but it seems like the authors have essentially added 6 new experiments in the rebuttal phase which I am not certain that they can incorporate in the main paper clearly. I have increased my score but still do not believe that this is a strong accept from my end.

**Key Questions For Authors:**

Apart from the concerns raised above that impact my scores for this paper, following are some general questions that have some implicit impact on my score:

1. What is the motivation behind the selection of the judge models (GPT-5/GPT-5.2 and GPT-5-mini)?
2. Is there a reason for not showcasing human qualitative analysis of LLM judgements? This will immediately show the benefits of the framework

**Limitations:**

Yes

**Strengths And Weaknesses:**

Strengths:
- The definitions for the metrics are well grounded and useful for effective evaluation.
- The per component ablation is very useful and shows clear improvements with each of dimensions, granular rubrics and expert guidance.
- The human judgment involved in curation of the dataset helps ground it in realism and is much needed for today's benchmarks that tend to be ecologically invalid.

Weaknesses
- The selection of judge models, specifically GPT-5.2 and GPT-5-mini is unclear. Ablations with different judge backbones are needed to prove the effectiveness of the evaluation framework.
- Human performance is not reported in Table 1 which gives the reader no comparison point for performance.
- The selection of a 1-10 over a 1-5 scale for judgment is unclear. According to Table 12, it seems like the scoring scale of 1-10 is collapsed into 5 categories of judgment anyway.
- With reference to Table 2, two expert judgments are not enough to show significance for deepresearch tasks that tend to be subjective on multiple metrics used in the paper. Furthermore, according to Appendix H, the humans are asked to rate on a scale of 1-5 but the models rate at a different scale of 1-10. The rescaling of human ratings in this manner leads to loss of specificity.
- The LLM judge evaluates the entire deepresearch output in one shot. Even though the expert guidance helps grounding, previous works such as [1] have studied that LLM judges have length biases. No study has been done on the lengths of the reports produced and comparing these against the LLM judge performance.
- The requirement of expert guidance for evaluation makes the method difficult to scale and generalize.
- Table 4 compares GPT-5 against GPT-5-mini but according to line 270, the correct model to be tested here must be GPT-5.2 instead.


[1] Domhan, Tobias, and Dawei Zhu. "Same evaluation, more tokens: On the effect of input length for machine translation evaluation using Large Language Models." Proceedings of the 2025 Conference on Empirical Methods in Natural Language Processing. 2025.

---

> ### Author Rebuttal · Authors · 2026-03-31
>
> We sincerely thank the reviewer for the thoughtful and constructive feedback.
> ### W1, Q1. Judge selection is unclear
> We selected the judge models based on both performance and cost efficiency. The tables below present judge-backbone ablations.
>
> **Report Quality Judge – Human Correlation and Cost**
>
> | Model | Pearson | Spearman | Pairwise | Cost In | Cost Out |
> | - | - | - | - | - | - |
> | GPT-5.2 | 0.86 | 0.80 | 0.89 | 1.75 | 14.0 |
> | Gemini 2.5 Pro | 0.76 | 0.70 | 0.82 | 1.25–2.50 | 10.0–15.0 |
> | GPT-5-mini | 0.75 | 0.73 | 0.87 | 0.25 | 2.00 |
> | Claude Opus 4.1 | 0.63 | 0.67 | 0.82 | 15.0 | 75.0 |
> | Claude Sonnet 4.5 | 0.73 | 0.63 | 0.80 | 3.00 | 15.0 |
>
> **Information Verification – Performance and Cost**
>
> Claim Extraction / Classification
> | Model | Recall | Cls. F1 | Cost |
> | - | - | - | - |
> | GPT-5-mini | 92.1 | 68.8 | 0.16 |
> | GPT-5 | 89.2 | 66.0 | 0.59 |
> | Claude Haiku 4.5 | 91.9 | 65.8 | 1.45 |
> | Gemini 2.5 Flash | 91.2 | 60.1 | 0.44 |
>
> Claim Verification
> | Model | F1 | Cost |
> | - | - | - |
> | GPT-5-mini | 87.3 | 0.95 |
> | Gemini 2.5 Flash | 86.3 | 1.28 |
> | Claude Haiku 4.5 | 86.7 | 2.87 |
>
> For Report Quality, we use GPT-5.2 for its best overall performance-cost trade-off. For Information Verification, each claim must be checked against external documents, which requires a large number of API calls, so we use GPT-5-mini for its strong performance at much lower cost. We will clarify this rationale in the final version.
>
> ### W2. lacks human performance..
> Human performance would be a useful comparison point, but collecting expert-written reports at scale is prohibitively expensive: 50 expert guidance annotations alone cost about 5k USD, and adding expert-written reports would require at least an additional 20k USD. Recent deep research benchmarks [1,2] likewise typically omit such baselines.
>
> ### W3. The motivation for using a 1–10 scale..
> The five score bands serve as anchors, while the 1–10 scale is used to capture finer-grained differences within each band. In addition, on a 10-point scale, a one-point scoring error has a smaller relative effect than on a 5-point scale. In contrast, a 5-point scale is sufficient for humans, who can clearly distinguish among the five categories.
>
> ### W4. Two expert ratings are insufficient & Rescaling human 1–5 ratings to 1–10 may lose specificity.
> The human evaluation in Table 2 does not rely on the judgments of the same two raters across all tasks. Instead, each of the 15 tasks was evaluated by a different pair of domain experts for that specific topic. Thus, it is not based on a small fixed set of raters, but on distributed evaluations from multiple experts across topics. We also observe relatively high inter-human agreement, with a pairwise agreement of 0.79. Therefore, we believe the human evaluation in Table 2 provides a meaningful comparison point grounded in the consensus of multiple experts.
>
> In addition, Appendix H does not rescale human ratings, but model scores, by dividing the 1–10 scores by 2. Since this is a linear transformation, it does not materially affect comparison metrics such as Pearson, Spearman, or pairwise agreement, so it is difficult to argue that specificity is lost.
>
> ### W5. LLM judge length bias.
> To examine this concern, we analyzed the correlation between report length (in words) and evaluation score across all 800 baseline reports. We found that the Pearson correlation between report length and the average score aggregated over the five report-quality dimensions was -0.0153, and the Spearman correlation was 0.1003, indicating almost no overall correlation. This suggests that the evaluation results are unlikely to be driven simply by longer outputs. We will include this analysis as a table in the final version.
> | Metric | Pearson | p | Spearman | p |
> | - | - | - | - | - |
> | Avg. Score | -0.02 | 0.67 | 0.10 | <0.01 |
>
> ### W6. Requiring expert guidance limits scalability and generalizability.
> We believe this concern stems from viewing expert guidance as part of a general-purpose evaluation methodology rather than as part of benchmark design. Our goal is not to propose a universally scalable evaluation method, but to build a benchmark that assesses deep research capability with high reliability and validity. In this context, the key question is not whether the methodology itself is universally scalable, but whether expert guidance improves the benchmark’s reliability and validity.
>
> ### W7.  Table 4  the correct model must be GPT-5.2
> As noted in Appendix G, we use GPT-5 as a shorthand in the paper. However, to avoid confusion, we will revise Table 4 and related references to consistently use GPT-5.2.
>
> ### Q2. qualitative analysis
> We agree and will add a qualitative comparison of human and LLM evaluations in the final version to highlight the benefits of Granular Rubrics and Expert Guidance; examples are omitted in the rebuttal due to space limits.
>
> [1] ResearchRubrics ICLR 2026.
> [2] LiveResearchBench ICLR 2026.

---

> > ### Author Rebuttal · Reviewer_fGXM · 2026-03-31
> >
> > 1. Details of the human-judge correlation study are not provided in the rebuttal and so validation of these scores is difficult
> > 2. The use of 1-10 is still ungrounded and a finer rubric should be defined. I disagree that the linear transformation is simply scaling since this heavily depends on said rubric. Most rubrics do not scale linearly and it is really hard to understand what makes a rating of, for example, 5 different from a 7 in open ended settings. The authors heavily rely on this vaguely defined scale across the paper which makes me trust the results lesser.
> > 3. "We analyzed the correlation between report length (in words) and evaluation score across all 800 baseline reports." - this study is not meaningful to show that your judges are not biased by length. If longer reports in domain A score higher while longer reports in domain B score lower, those effects cancel out in the aggregate, producing a near-zero correlation even though length bias exists within subgroups. This is where human evaluation is important since such issues are only reliably flagged by human review.
> > 4. The lack of qualitative analysis is still problematic since we don't have any insight into the strengths and weaknesses of DEER.
> >
> > The fact that the whole framework is sensitive to LLMs (from open ended judges, lack of proper rubric and reliance on several API calls where attribution of evaluator errors is hard) has solidified my score here.

---

> > > ### Author Response · Authors · 2026-04-08
> > >
> > > We thank the reviewers for the clarification.
> > >
> > > ## 1. Details of the human-judge correlation
> > > The human-correlation results added in our rebuttal for judge-backbone comparison follow the Table 2 / Appendix H setup: 45 reports, two domain-expert ratings per report, and the same Pearson/Spearman/pairwise-agreement computation.
> > >
> > > ## 2. 1-10 is still ungrounded
> > > The reviewer’s concern treats our 1–10 scale as an unanchored 10-point free-form rating. That is not the case. As defined in Table 12, our scoring is grounded in five bands with descriptions: 1–2 Poor, 3–4 Inadequate, 5–6 Good, 7–8 Excellent, and 9–10 Perfect. Thus, the 1–10 format is not ten unrelated score categories, but a higher-resolution version of this five-band scale. For example, the reviewer’s scores of 5 and 7 correspond to the Good and Excellent bands, respectively.
> > >
> > > Likewise, the 1–10 to 1–5 mapping in Appendix H is not an arbitrary score compression. Because the scale is defined by the same five bands, the mapping preserves that structure while changing only its numerical range. For example, 9 and 10 both remain in the Perfect band but are still distinguished after conversion as 4.5 and 5.
> > >
> > > This transformation does not distort the human-correlation results in Table 2: Pearson is invariant to positive linear transformations, while Spearman correlation and pairwise agreement depend on relative ordering. Therefore, this transformation does not artificially increase or decrease human alignment.
> > >
> > > Moreover, under the same Table 2 / Appendix H setup—the 45 reports, two domain-expert ratings, the mean over the five evaluators, and the same metrics—the 5-band + 10 scale design shows higher human alignment on all three metrics than both 5-band + 5 scale and no-anchor + 10 scale. This shows that our scale is grounded and empirically better aligned with expert judgment.
> > >
> > > | Setting | Pearson | Spearman | Pairwise |
> > > |-|-|-|-|
> > > | 5-band anchor + 10 scale | 0.75 | 0.71 | 0.84 |
> > > | 5-band anchor + 5 scale | 0.70 | 0.67 | 0.82 |
> > > | No-anchor + 10 scale | 0.73 | 0.65 | 0.80 |
> > >
> > > ## 3. length bias exists within subgroups
> > > We directly tested that possibility at the task level. Most task-level correlations between report length and evaluation score are near zero, and the few larger correlations are rare and inconsistent in direction. Thus, the near-zero aggregate correlation is unlikely to result from hidden subgroup effects.
> > >
> > > Task-level correlation between length and score (50 tasks)
> > > | Correlation range | Pearson (# tasks) | Spearman (# tasks) |
> > > |---|---|---|
> > > | [-0.6, -0.4) | 1 | 1 |
> > > | [-0.4, -0.2) | 3 | 7 |
> > > | [-0.2, 0.2) | 36 | 32 |
> > > | [0.2, 0.4) | 6 | 7 |
> > > | [0.4, 0.6) | 4 | 3 |
> > >
> > > ## 4. Qualitative analysis
> > > In response, we added a qualitative analysis. Due to space limitations, we present only one representative analysis for the task in Figure 5. Specifically, we compared the evaluation results for the same report when only the 101 rubric was provided versus when Expert Guidance (EG) was provided together, and examined how the results differ.
> > >
> > > | Rubric Item | 101 rubric | 101 rubric + EG |
> > > |-|-|-|
> > > | 1.1.1.1 Include all elements required by the User Query | 10 points (Perfect) — judged to include all required elements | 6 points (Good) — The convergence-rate analysis is present, but it lacks refined error bounds and comparisons across edge cases. Explicit counterexamples, practical validation, and reproducible examples are also limited, and key equations necessary for the convergence guarantee argument are missing. Therefore, it falls under Good, as it covers more than half of the requirements. |
> > >
> > > By directly reviewing the report, we found that although it satisfied much of the requirements, several key elements were still missing, including the contraction mapping proof, geometric convergence bounds, analysis of reward sign and symmetry, and the design of reproducible examples. This confirms that EG enables more accurate evaluation. In the final version, we will add analyses of diverse cases.
> > >
> > > ## 5. Sensitive to LLMs
> > > Characterizing our framework as open-ended judging without a proper rubric does not fully reflect the core design of our method. DEER performs evaluation based on 101 rubric items and task-specific Expert Guidance.
> > >
> > > Importantly, we directly validated the validity and consistency of our method in the paper. Table 2 shows that, across multiple evaluator models, DEER’s scores align with expert human judgment, and Table 3 shows that the results remain consistent across evaluators. Thus, the reviewer’s concern about LLM sensitivity is directly addressed in terms of both human alignment and inter-evaluator reliability.
> > >
> > > If the concern instead relates to the reliance on several API calls in the information-verification pipeline, the paper evaluates each stage separately: claim extraction (Table 4), backtracking (Table 15), and verification (Table 5). Therefore, evaluator errors are not opaque end-to-end failures, but can be inspected at the component level.

---

### Official Review · Reviewer_wGTD · 2026-03-10

**Soundness:** 3
**Presentation:** 2
**Significance:** 4
**Originality:** 3
**Overall Recommendation:** 4
**Confidence:** 3

**Summary:**

This paper introduces DEER, a new benchmark designed to evaluate deep research agents on generating expert-level reports. The authors created a highly detailed evaluation framework with 101 rubric items and built a claim verification system that checks even uncited claims against external evidence. They used this toolkit to test several baseline models, revealing a clear picture of where current systems excel and where they still fall short in professional writing.

**Compliance With Llm Reviewing Policy:**

Affirmed.

**Key Questions For Authors:**

In your experiments with the baseline models (like OpenAI, Claude, WebThinker), the models use their own built-in web search systems. Doesn't this mean the retrieval sources and search mechanisms are completely different across platforms? If so, how do you handle or measure the variance caused by these different retrieval pipelines versus the actual reasoning and writing capabilities of the models themselves?

**Limitations:**

yes

**Strengths And Weaknesses:**

The core idea is fantastic. Using such a fine-grained evaluation taxonomy (7 dimensions and 25 subdimensions) provides a very thorough and complete way to measure report quality.

Evaluating long-form reports is notoriously difficult, and the authors clearly put a ton of labor-intensive work into crafting the expert guidance and the evaluation pipeline. Releasing these resources will be super valuable for the research community.

The experimental results are genuinely interesting and deserve a close look.

My main suggestion is about the paper's structure. Right now, a lot of the heavy lifting is done in the appendices. I highly recommend moving some of the more detailed design considerations and core methodology (like a condensed version of the rubric items or examples of the semantic back-tracking) into the main text. This would make the paper much easier to follow and significantly reduce the reader's burden.

---

> ### Author Rebuttal · Authors · 2026-03-31
>
> ### W. My main suggestion is about the paper's structure. Right now, a lot of the heavy lifting is done in the appendices. I highly recommend moving some of the more detailed design considerations and core methodology (like a condensed version of the rubric items or examples of the semantic back-tracking) into the main text. This would make the paper much easier to follow and significantly reduce the reader's burden.
>
> Thank you for this valuable suggestion. We agree that some important design considerations and parts of the core methodology are currently placed in the appendix, which may make the paper harder to follow. Accordingly, in the final version, we will move a condensed summary of the rubric items, concrete examples of semantic back-tracking, and other key methodological details into the main text to improve readability and ease of understanding.
>
> ### Q. In your experiments with the baseline models (like OpenAI, Claude, WebThinker), the models use their own built-in web search systems. Doesn't this mean the retrieval sources and search mechanisms are completely different across platforms? If so, how do you handle or measure the variance caused by these different retrieval pipelines versus the actual reasoning and writing capabilities of the models themselves?
>
> Thank you for this thoughtful question. As you pointed out, platforms such as OpenAI, Claude, and WebThinker likely rely on different search sources and retrieval mechanisms. However, because these search pipelines are generally not publicly disclosed and are not accessible in a controllable form for researchers, our benchmark does not directly disentangle variance arising from the search pipeline from differences in the model’s own reasoning and writing abilities. This is a limitation shared by prior benchmarks as well.
>
> That said, while we cannot fully separate these factors, our multidimensional evaluation framework allows us to assess them indirectly. The **Information Sufficiency** and **Information Integrity** dimensions, which are tied to information verification, partially reflect the quality of the search pipeline. For example, **Information Sufficiency** evaluates whether the model gathers and uses sufficient information through sub-metrics such as **Evidence Coverage, Claim Amount, Citation Amount, and Reference Amount**. Similarly, **Information Integrity** assesses whether the retrieved information is factually sound and properly supported through metrics such as **Claim Factuality, Citation Support, Reference Support, Reference Quality, and Reference Diversity**.
>
> By contrast, the dimension more closely related to the model’s own capabilities is **Report Quality**, which can be further interpreted in terms of **r**easoning ability and writing ability. Reasoning ability is captured by **Request Fulfillment** (**Completeness, Scope, Helpfulness**) and **Analytical Soundness** (**Quantification, Reasoning**), which reflect how well the model addresses the user’s request and how logically and analytically it interprets the collected information. Writing ability, on the other hand, is reflected in **Structural Coherence** (**Introduction, Body, Conclusion, Section**) and **Format & Style** (**Report Format, Writing Quality, Paragraph Quality, Readability**), which evaluate how well the model organizes information into a report and how polished and readable the resulting writing is. In addition, **Ethics & Compliance** assesses whether the generated report satisfies safety and normative standards.
>
> Indeed, as shown in Table 1, when search is added to a reasoning-only baseline, performance on the Report Quality dimension remains largely stable, while Information Sufficiency and Information Integrity improve. This suggests that adding search primarily strengthens information gathering and verification, while the model’s underlying reasoning and writing quality remains relatively stable.
>
> Therefore, rather than claiming that we fully disentangle the effects of the search pipeline from the model’s intrinsic capabilities, it is more accurate to say that our evaluation framework decomposes performance factors that were entangled in prior benchmarks, making it possible to examine the influence of the search pipeline as well as the model’s reasoning and writing abilities in a more fine-grained way.

---

> > ### Author Rebuttal · Reviewer_wGTD · 2026-04-01
> >
> > I think the current overall recommendation is appropriate.

---

> > > ### Author Response · Authors · 2026-04-08
> > >
> > > We sincerely appreciate your time and effort in reviewing our paper.

---

### Official Review · Reviewer_W6vr · 2026-03-10

**Soundness:** 3
**Presentation:** 3
**Significance:** 3
**Originality:** 2
**Overall Recommendation:** 5
**Confidence:** 1

**Summary:**

This paper introduces DEER, a benchmark for evaluating AI systems that generate expert-style long-form research reports. Instead of relying on a single overall score, DEER evaluates reports along multiple dimensions of quality, including whether the report fulfills the user’s request, is analytically sound, is well structured, and is properly grounded in evidence. To support this, the authors build 50 report-generation tasks across 13 domains and pair them with expert-designed evaluation guidance and fine-grained rubrics.

A second contribution is the benchmark’s information-verification pipeline. The framework extracts claims from reports, identifies which ones are verifiable, and checks whether they are supported by cited evidence, including support that may come indirectly from earlier context rather than only from the local sentence. Using this evaluation setup, the paper studies current deep-research systems and finds that they are generally stronger on presentation-related aspects of report writing than on request fulfillment and analytical depth. Overall, the paper’s main contribution is a more diagnostic and expert-grounded evaluation framework for long-form research agents.

**Compliance With Llm Reviewing Policy:**

Affirmed.

**Key Questions For Authors:**

1. How robust is DEER to errors in the evaluation pipeline itself, especially claim extraction, claim typing, and citation backtracking? If you can show these intermediate steps are reliable, it would increase my confidence in the benchmark’s soundness.

2. Which components matter most for human alignment: fine-grained rubrics, expert evaluation guidance, or the information-verification module? A clearer breakdown would help me better understand the source of the gains and could affect my assessment of originality and soundness.

3. How sensitive are the system rankings to the choice of judge model or verification model used inside DEER? If rankings are stable across evaluator choices, that would make the benchmark substantially more convincing.

**Limitations:**

Yes

**Strengths And Weaknesses:**

Soundness: The paper is fairly solid for an evaluation-benchmark submission. The authors identify a real gap in current assessment of deep-research agents: existing evaluations are often too holistic or too shallow to distinguish polished writing from genuinely expert, well-supported analysis. Their response is methodologically appropriate for that problem. DEER combines a rubric-based report-quality framework with a claim-verification pipeline, and the paper validates the evaluator by comparing several judging setups against expert human assessments. A clear strength is that the authors do not just assert that their rubric works; they show that adding task-specific expert guidance substantially improves agreement with humans, which makes the evaluation pipeline more credible.

The empirical design is also reasonably thoughtful. The benchmark spans 50 tasks across 13 domains, and the authors analyze multiple systems rather than just showcasing a single example. The separation between report-quality evaluation and evidence verification is another sound design choice, since those are genuinely distinct failure modes. The claim-level verification pipeline, including backtracking for implicitly supported claims, is a meaningful attempt to move beyond naive citation checking.

The main weakness on soundness is that, like many LLM-as-a-judge papers, the benchmark still depends heavily on the quality of the evaluation pipeline itself. Although the human-alignment analysis is useful, the benchmark ultimately relies on another stack of model-based extraction, classification, and judging decisions. That does not make the paper unsound, but it does mean the conclusions should be interpreted as “credible under this evaluation framework,” not as ground truth. Relatedly, the task set is high quality but still modest in size, so some caution is warranted about broad generalization across all deep-research use cases.

Presentation: The paper is well written and well structured. The motivation is easy to follow, and the decomposition into report quality versus information verification gives the work a clean narrative. The rubric taxonomy is clearly organized, and the paper does a good job explaining why simple holistic scoring is insufficient for long-form expert reports. I also found the validation story coherent: first define the benchmark dimensions, then build the scoring pipeline, then test how well it matches experts, and finally use it to analyze current systems.

A presentation strength is that the paper is quite diagnostic. It does not stop at saying one system is better than another; it uses the benchmark dimensions to show where current systems are strong and weak. That makes the narrative more informative than a pure leaderboard paper.

The main presentation weakness is complexity. There are many moving parts: rubric construction, expert guidance, claim extraction, claim typing, citation backtracking, verification metrics, and aggregation. Each piece is motivated, but the end-to-end pipeline is somewhat elaborate, and it can be hard to tell which pieces are essential versus auxiliary. The paper would be even clearer with a more explicit summary of the minimum core ingredients needed to reproduce the benchmark faithfully.

Significance: I think the paper addresses an important problem. As more systems are marketed as deep-research or expert-report agents, evaluation quality becomes a bottleneck. A report that looks polished but is poorly reasoned or weakly supported is exactly the kind of failure that matters in practice. So a benchmark that explicitly measures request fulfillment, analytical soundness, and evidence grounding is relevant and useful. This is especially timely because long-form agent outputs are becoming more common, while evaluation standards remain underdeveloped.

The likely impact is meaningful. Even if the benchmark is not huge, it could influence how future research papers evaluate long-form research agents, moving the field away from vague preference judgments and toward more diagnostic assessment. The benchmark also yields a useful substantive finding: current systems appear stronger on style and formatting than on true request fulfillment and analytical depth. That kind of insight is valuable for both research and product development.

The main limitation on significance is scope. DEER is specialized to expert-style report generation, not general LLM evaluation, so its impact will likely be strongest in the growing but still specific area of long-form research agents. That said, this specialization feels appropriate rather than narrow in a bad way, because the benchmark is tailored to a real emerging use case.

Originality: The paper’s originality is moderate but meaningful. It does not introduce a new model or a new learning algorithm. Its novelty is in the benchmark design and evaluation methodology: combining a domain-informed rubric hierarchy, task-specific expert evaluation guidance, and a report-wide claim-verification procedure that accounts for both explicitly and implicitly supported claims. That is a thoughtful synthesis rather than a trivial recombination.

I also think the paper contributes a useful perspective: long-form report evaluation should separate writing quality from information support, rather than collapsing them into one holistic judgment. That framing is not radically new in spirit, but the paper operationalizes it more concretely than most prior evaluation work.

The limitation is that the originality is mostly methodological and organizational, not conceptual in a deep theoretical sense. Many ingredients individually have clear precedents, including rubric-based judging, LLM-as-a-judge setups, and citation verification. So I would not describe this as a major conceptual breakthrough. Instead, I would describe it as a careful and useful benchmark contribution whose originality lies in the integration and task-specific adaptation of existing ideas.

Overall, I view the paper as strongest on significance and practical usefulness. It identifies a real evaluation gap, proposes a reasonably well-validated benchmark, and yields insights that seem actionable. Its main weaknesses are the inherent dependence on a complex evaluator stack and the limited task scale, but for a benchmark paper the work appears thoughtful, timely, and technically credible.

---

> ### Author Rebuttal · Authors · 2026-03-31
>
> We greatly appreciate the reviewer’s careful reading and insightful feedback
> ### Q1. How robust is DEER...
>
> Each stage of the Information Verification pipeline is individually validated against expert-annotated ground truth. For claim extraction, we measure the semantic equivalence between model-extracted claims and ground truth, achieving a high recall of 92.17% (Table 4). For claim classification, we achieve a 6-class F1 of 68.8; additionally, our analysis shows that the operationally critical metric — binary verifiable (A–C) vs. non-verifiable (D–F) classification — reaches F1 79, confirming that claims requiring external evidence are reliably captured while unnecessary claims are safely excluded (Table 4). For citation back-tracking, we validate by comparing LLM-predicted citations propagated from dependent sentences against ground truth citations, achieving a superior precision-Jaccard balance over sliding window baselines (Appendix F.2, Table 15).
>
> From a coverage perspective, only 62.5% of all claims requiring verification carry explicit citations. Without backtracking, at most 62.5% of verifiable content can be assessed; with backtracking, even accounting for prediction errors, coverage extends to approximately 91%. Taken together, these results demonstrate that DEER's Information Verification pipeline is robust across all validated stages, and we will include these analyses explicitly in §6.5 of the revised version.
>
>
> ### Q2. Which components matter most for human alignment...
>
> We validated human alignment from two complementary perspectives: agreement with a rubric-based LLM judge for report quality, and performance on information verification. For the latter, because holistic human evaluation of both the report and its supporting sources is difficult, subjective, and time-consuming, we instead used a separate human-constructed test set to assess each module.
>
> Meanwhile, as shown in Table 2, the factor that contributed most to human alignment in the Report Quality LLM judge was the expert evaluation guidance. Fine-grained rubrics have the advantage of providing detailed and systematic evaluation criteria, but when used alone, they actually showed a tendency to reduce alignment with human judgments. This suggests that even when detailed rubrics are available, it is still difficult for the LLM judge to appropriately identify and evaluate all of the key factors.
>
> By contrast, when task-specific expert evaluation guidance was provided, the model was able to more clearly refer to the key elements and details that should be considered important in each task, grounded in the relevant professional context. As a result, the LLM judge was better able to follow the judgment criteria used by human evaluators. In other words, if fine-grained rubrics provide the overall structure of evaluation, expert evaluation guidance indicates which factors should be interpreted as more important in the context of each task or domain, and this appears to have contributed substantially to improving human alignment.
>
> ### Q3. How sensitive are the system rankings…
> The consistency of evaluation results across different judge models can be seen in Table 3. As shown there, our final proposed method achieves the best performance on Krippendorff’s α and ICC, indicating the highest inter-evaluator reliability among the compared settings.
>
> However, this result reflects only score consistency and does not directly address the stability of system rankings raised in the reviewer’s question. We therefore conducted an additional analysis under the same setting as Table 3 to measure how consistently different evaluator models preserve the relative rankings of systems. Specifically, based on the scores assigned by each evaluator to the same set of reports, we derived system-level rankings and measured ranking agreement using Kendall’s W, average pairwise Spearman’s ρ, and average pairwise Kendall’s τ.
>
> The results show that our final proposed method also achieves the strongest agreement from the ranking perspective. This suggests that our method is robust to evaluator model choice in terms of the stability of system rankings.
>
> | Method | Kendall’s W | Avg Pair Spearman’s ρ | Avg Pair Kendall’s τ |
> |-|-|-|-|
> | Vanilla | 0.66 | 0.58 | 0.53 |
> | + Dimensions | 0.66 | 0.58 | 0.49 |
> | + Granular Rubrics | 0.58 | 0.48 | 0.43 |
> | + Expert Guidance | 0.74 | 0.68 | 0.56 |
>
> Furthermore, we verified the robustness of our Information Verification module by substituting our primary judge (gpt-5-mini) with two alternative models (gemini-2.5-flash and claude-haiku-4.5) across research reports. As shown below, cross-evaluator ranking agreement remained exceptionally high for both Integrity and Sufficiency metrics (Kendall’s W > 0.93 and Avg Pairwise Spearman’s ρ > 0.90).
>
> | Dimension | Kendall’s W | Avg Pair Spearman’s ρ | Avg Pair Kendall’s τ |
> |-|-|-|-|
> | Integrity | 0.93 | 0.90 | 0.74 |
> | Sufficiency | 0.93 | 0.90 | 0.74 |

---

> > ### Author Rebuttal · Reviewer_W6vr · 2026-04-03
> >
> > Thanks for the explanation. My concerns have been partially addressed.

---

> > > ### Author Response · Authors · 2026-04-08
> > >
> > > Thank you again for your careful reading and thoughtful feedback.
> > > To further address Q2, we conducted an additional ablation study. As clarified in the Reply Rebuttal Comment by Authors to Reviewer 3mTt, the added results indicate that both components contribute, but expert guidance seems to be the more important of the two for human alignment.

---

### Official Review · Reviewer_3mTt · 2026-03-13

**Soundness:** 3
**Presentation:** 3
**Significance:** 2
**Originality:** 2
**Overall Recommendation:** 4
**Confidence:** 4

**Summary:**

This paper presents DEER, a benchmark for evaluating deep research agents that generate expert-style reports for complex queries. The benchmark contains 50 report-generation tasks across 13 domains, derived from real user queries and expert-written questions. To evaluate reports, the authors define a taxonomy with 7 dimensions and 25 subdimensions (implemented as 101 rubric items) covering things like request fulfillment, analytical soundness, structure, style, and information use. They also provide task-specific expert guidance to help LLM judges evaluate reports more reliably. In addition to rubric-based scoring, DEER includes a claim verification module that extracts claims from the report, links both cited and uncited claims to evidence (using a backtracking method to recover missing citations), and checks whether the evidence supports the claims to measure information integrity and sufficiency. Experiments show that current deep research systems produce well-structured reports with citations, but still struggle to fully address the task and achieve strong analytical soundness.

**Compliance With Llm Reviewing Policy:**

Affirmed.

**Final Justification:**

The paper’s strengths are the structured evaluation taxonomy with 101 rubric items, the claim verification module that checks both cited and uncited claims, and the use of task-specific expert guidance to support more reliable evaluation. My main concerns were whether the HLE-derived tasks reflect real deep-research workflows at scale, whether the shared 101-item rubric adds clear value beyond task-specific guidance given the ablation results, and the initial lack of validation for the citation back-tracking mechanism.

The rebuttal clarifies the construction of report-style prompts from HLE seeds and strengthens confidence in the citation back-tracking pipeline through additional validation details. Overall, my recommendation is Weak Accept.

**Key Questions For Authors:**

Please see weaknesses.

**Limitations:**

yes

**Strengths And Weaknesses:**

## Strengths

1. DEER operationalizes research report quality into a structured hierarchy (with 101 rubric items), enabling detailed and interpretable evaluation signals.
2. Its information-verification module extracts claims, recovers implicit citations, and checks evidence support across the entire report rather than only explicitly cited sentences.
3. Task-specific guidance enumerates mandatory content elements so that LLM judges can detect domain-specific omissions and reasoning errors more reliably.

## Weaknesses

1. **\[Mismatch between exam-style prompts and real research workflows\]** The benchmark tasks are derived from Humanity's Last Exam questions, which were originally designed for short-answer problem solving. Although the authors rewrite these questions into report-style prompts through expert review, the underlying tasks may still retain an answer-oriented structure that differs from the open-ended and exploratory nature of real research workflows.
2. **\[Questionable benefit of scaling the shared rubric\]** The ablation results in Table 2 suggest that expert guidance contributes more to evaluation reliability than the granular shared rubric itself. In fact, adding granular rubrics alone reduces correlation with human judgments, while expert guidance produces the largest improvement. This raises the question of whether scaling the rubric to 101 shared items provides meaningful benefit beyond task-specific guidance. Moreover, expert guidance is conceptually similar to prior approaches that provide prompt-specific evaluation criteria or rubrics for LLM judges.
3. **\[Unvalidated citation back-tracking for uncited claims\]** The paper proposes citation back-tracking to verify uncited claims by propagating citations from semantically related prior sentences and presents this as a key contribution. But the reliability of this mechanism is not directly validated, and incorrect dependency predictions could cause unsupported claims to be incorrectly verified.

---

> ### Author Rebuttal · Authors · 2026-03-31
>
> Thank you for the thoughtful comments. We appreciate the reviewer’s careful reading of our paper and the constructive feedback.
>
> ### 1. [Mismatch between exam-style prompts and real research workflows]
>
> We understand the reviewer’s concern that, because the HLE seeds were originally in a QA format, the reformulated DEER tasks may still retain an answer-oriented structure. However, this is not how DEER queries were constructed. As described in Appendix B.3, we did not simply expand the original questions or answers; instead, we identified the underlying concepts, theories, and phenomena and reformulated them into research-oriented task queries for a deep research setting. We also removed “specific answers” from the prompt and instead required expert-style analysis, comparison, discussion of key issues, and reasoning/exposition. Cross-review further ensured that the tasks did not remain narrow answer-retrieval problems.
>
> This is also illustrated by the example in Appendix B.3. The MDP/value iteration task in Figure 5 does not ask for a single correct answer, but for a rigorous technical report covering convergence conditions, the roles of rewards and discounting, success and failure cases, practical factors, limitations, and open questions. Thus, the query itself is structured as a research task requiring multidimensional analysis and exposition.
>
> We agree that DEER does not fully capture the open-ended and exploratory nature of real-world research workflows, such as hypothesis generation or experiment design and execution. However, this is due to a difference in evaluation scope, not a flaw in the benchmark design. DEER is intended to evaluate the query-to-report setting commonly targeted by current deep research systems [1] and benchmarks [2,3]—retrieving, analyzing, synthesizing, and reporting on complex topics—rather than the entirety of real-world research.
>
> ### 2. [Questionable benefit of scaling the shared rubric]
> Although the reviewer is correct that adding the fine-grained shared rubrics alone reduced correlation with human judgment, this should not be interpreted as evidence that the 101 shared rubrics are not useful. Their purpose is not to maximize correlation, but to ensure broad coverage of key evaluation dimensions, reduce arbitrary judgments by the LLM judge, and provide interpretable, standardized axes for cross-task comparison. If correlation optimization were the goal, it would have been more natural to keep only the subset of criteria that aligned best with human judgments.
>
> Accordingly, Table 2 does not show that shared rubric expansion is ineffective. Rather, it shows that as evaluation criteria become broader and more fine-grained, existing LLM judges cannot assess them reliably on their own. This motivates our final design: combining a shared fine-grained rubric with task-specific expert guidance. The shared rubric provides coverage, interpretability, and cross-task comparability, while the expert guidance supplies the task-specific context needed for valid assessment.
>
> Additionally, our contribution is not simply using task-specific guidance, but constructing human-authored evaluation criteria refined by experts. This addresses concerns raised about AI-generated rubrics, including circularity and limited oversight [2], as well as their gap from human-annotated rubrics [4], and improves the validity and reliability of report evaluation.
>
> ### 3. [Unvalidated citation back-tracking for uncited claims]
> We have validated the reliability of each stage in the Citation Back-tracking pipeline against expert-annotated ground truth. The claim extraction and classification results reported in §6.5 and Table 4 confirm that the system reliably captures claims requiring external evidence while safely excluding those that do not. The back-tracking (dependency prediction) stage is directly compared against a sliding window baseline[5] in Appendix F.2 (Table 15), achieving a superior precision-Jaccard balance. We will add explicit forward references to these results in §6.5 of the revised version.
>
> Furthermore, our pipeline is designed to handle mispredicted dependencies safely. In the Claim Verification stage, even if an incorrect citation is inherited through back-tracking, the claim is marked as not supported if the cited source does not actually support it. Critically, in our analysis of the full pipeline against ground truth, we did not observe a single case in which an unsupported claim was incorrectly verified as supported.
>
> [1] OpenAI. “Introducing deep research.” 2, 2, 2025.
> [2] ResearchRubrics: A Benchmark of Prompts and Rubrics for Evaluating Deep Research Agents. ICLR 2026.
> [3] LiveResearchBench: A Live Benchmark for User-Centric Deep Research in the Wild. ICLR 2026.
> [4] RubricBench: Aligning Model-Generated Rubrics with Human Standards. arXiv 2026.
> [5] Deepscholar-bench: A live benchmark and automated evaluation for generative research synthesis. arXiv 2025.

---

> > ### Author Rebuttal · Reviewer_3mTt · 2026-04-03
> >
> > Thank you for addressing my comments. The rebuttal clarifies the construction of report-style prompts from HLE seeds and strengthens confidence in the citation back-tracking pipeline through additional validation details. However, two concerns still remain:
> >
> > - It is still unclear whether the resulting tasks adequately reflect the open-ended structure of real deep-research workflows at scale.
> > - The marginal benefit of the shared 101-item rubric remains unclear because the ablations suggest that task-specific expert guidance, not rubric granularity, is what primarily improves agreement with human judgments.
> >
> > I will maintain my current score.

---

> > > ### Author Response · Authors · 2026-04-08
> > >
> > > We appreciate the reviewers’ feedback on the parts that were still unclear.
> > >
> > > ### **It is still unclear whether the resulting tasks adequately reflect the open-ended structure of real deep-research workflows at scale.**
> > >
> > > We understand this comment as raising the question of whether DEER’s tasks sufficiently reflect the open-ended and exploratory structure of real deep-research workflows. To address this point, it is first necessary to distinguish the notion of *deep research* used in current systems and benchmarks from open scientific research. In the existing deep-research literature and systems, *deep research* is generally understood as a workflow that explores external information in response to a complex user query, analyzes and synthesizes that information, and organizes the results into an evidence-grounded long-form report. For example, OpenAI describes deep research as a reasoning, research, and synthesis workflow for documented reports [1]. Representative benchmarks and systems are broadly consistent with this framing: Mind2Web2[2] evaluates deep-research agents primarily in terms of search and browsing, while DeepResearchBench[3], LiveResearchBench[4], and ResearchRubrics[5] evaluate systems along the axis of web exploration, synthesis, and report generation. WebThinker[6] and WebWeaver[7] are likewise situated in evaluation settings centered on web exploration, synthesis, and report generation. In other words, the current deep-research literature does not primarily target the full process of scientific research itself, but rather a shared workflow of information seeking, analysis, synthesis, and report writing. DEER is designed to evaluate precisely this workflow.
> > >
> > > In this sense, we believe that DEER adequately reflects the open-ended structure of real deep-research workflows. DEER's tasks do not require simple fact retrieval or completion of a single reasoning path. Instead, they require the model to judge what information to seek on complex expert-level report topics, compare and organize evidence from diverse sources and perspectives, and structure the results into a coherent long-form report. There is no single predetermined answer format for these tasks — different expert-style reports are possible depending on what evidence is found and prioritized, what comparative axes are emphasized, and how the narrative structure is organized. In this sense, DEER's tasks are open-ended. They require information seeking, importance judgment, perspective integration, and structured exposition, which reflects the core structure of the query-to-report workflow that current deep research systems actually perform. This property is not specific to a single example but is common across all tasks.
> > >
> > > [1] OpenAI. “Introducing Deep Research.” February 2, 2025.
> > > [2] “Mind2Web 2: Evaluating Agentic Search with Agent-as-a-Judge.” NeurIPS 2025.
> > > [3] “DeepResearch Bench: A Comprehensive Benchmark for Deep Research Agents.” ICLR 2026.
> > > [4] “LiveResearchBench: A Live Benchmark for User-Centric Deep Research in the Wild.” ICLR 2026.
> > > [5] “ResearchRubrics: A. Benchmark of Prompts and Rubrics for Evaluating Deep Research Agents.” ICLR 2026.
> > > [6] “WebThinker: Empowering Large Reasoning Models with Deep Research Capability.” NeurIPS 2025.
> > > [7] “WebWeaver: Structuring Web-Scale Evidence with Dynamic Outlines for Open-Ended Deep Research.” ICLR 2026.
> > >
> > > ### **The marginal benefit of the shared 101-item rubric remains unclear**
> > > We agree that our prior rebuttal did not make the marginal benefit of the shared granular rubrics sufficiently explicit. To address this, we conducted an additional ablation comparing the full DEER evaluator against variants with either the shared granular rubrics or the task-specific expert guidance removed. The setup follows Table 2, and the newly added DEER − Granular Rubrics applies expert guidance only, without the 101 shared rubric items.
> > >
> > > | Evaluation Method | Pearson r | Spearman ρ | Pairwise |
> > > |---|---:|---:|---:|
> > > | DEER − Granular Rubrics | 0.663 | 0.648 | 0.782 |
> > > | DEER − Expert Guidance | 0.621 | 0.593 | 0.783 |
> > > | DEER | 0.746 | 0.706 | 0.840 |
> > >
> > > Removing the shared granular rubrics leads to a consistent drop in agreement with human judgment across all three metrics (Pearson 0.746→0.663, Spearman 0.706→0.648, Pairwise 0.840→0.782). This shows that the full performance cannot be attributed to expert guidance alone, and that the shared granular rubrics make a measurable independent contribution. We therefore interpret the two components as complementary rather than substitutable, and will incorporate this ablation into the revised version to clarify their respective roles.

---

### Decision · Program_Chairs · 2026-04-30

**Decision:**

Accept (regular)

**Comment:**

This paper introduces DEER, a multifaceted benchmark for reports generated by "deep research agents". There was consensus that this is a valuable contribution given the rise of deep report generation systems and the difficulty of evaluating the outputs that they generate. The system comparisons on offer (made using the benchmark) may also be of interest to the ICML community. Questions around benchmark construction and metric validity were satisfactorily addressed in rebuttal.